# Phytochemical and Potential Properties of Seaweeds and Their Recent Applications: A Review

**DOI:** 10.3390/md20060342

**Published:** 2022-05-24

**Authors:** Hossam S. El-Beltagi, Amal A. Mohamed, Heba I. Mohamed, Khaled M. A. Ramadan, Aminah A. Barqawi, Abdallah Tageldein Mansour

**Affiliations:** 1Agricultural Biotechnology Department, College of Agriculture and Food Sciences, King Faisal University, Al-Ahsa 31982, Saudi Arabia; 2Biochemistry Department, Faculty of Agriculture, Cairo University, Giza 12613, Egypt; 3Chemistry Department, Al-Leith University College, Umm Al-Qura University, Makkah 24831, Saudi Arabia; aabbarqawi@uqu.edu.sa; 4Plant Biochemistry Department, National Research Centre, Cairo 12622, Egypt; 5Biological and Geological Science Department, Faculty of Education, Ain Shams University, Cairo 11757, Egypt; 6Central Laboratories, Department of Chemistry, King Faisal University, Al-Ahsa 31982, Saudi Arabia; kramadan@kfu.edu.sa; 7Biochemistry Department, Faculty of Agriculture, Ain Shams University, Cairo 11566, Egypt; 8Animal and Fish Production Department, College of Agricultural and Food Sciences, King Faisal University, Al-Ahsa 31982, Saudi Arabia; amansour@kfu.edu.sa; 9Fish and Animal Production Department, Faculty of Agriculture (Saba Basha), Alexandria University, Alexandria 21531, Egypt

**Keywords:** antioxidant activity, functional foods, health benefits, seaweeds, secondary metabolites

## Abstract

Since ancient times, seaweeds have been employed as source of highly bioactive secondary metabolites that could act as key medicinal components. Furthermore, research into the biological activity of certain seaweed compounds has progressed significantly, with an emphasis on their composition and application for human and animal nutrition. Seaweeds have many uses: they are consumed as fodder, and have been used in medicines, cosmetics, energy, fertilizers, and industrial agar and alginate biosynthesis. The beneficial effects of seaweed are mostly due to the presence of minerals, vitamins, phenols, polysaccharides, and sterols, as well as several other bioactive compounds. These compounds seem to have antioxidant, anti-inflammatory, anti-cancer, antimicrobial, and anti-diabetic activities. Recent advances and limitations for seaweed bioactive as a nutraceutical in terms of bioavailability are explored in order to better comprehend their therapeutic development. To further understand the mechanism of action of seaweed chemicals, more research is needed as is an investigation into their potential usage in pharmaceutical companies and other applications, with the ultimate objective of developing sustainable and healthier products. The objective of this review is to collect information about the role of seaweeds on nutritional, pharmacological, industrial, and biochemical applications, as well as their impact on human health.

## 1. Introduction

Seaweeds have received lot of attention in recent years because of their incredible potential. Seaweeds are essential nutritional sources and traditional medicine components [1]. Marine macroalgae, sometimes known as seaweeds, are microscopic, multicellular, photosynthetic eukaryotic creatures. Based on their coloration and depending on their taxonomic classification, they can be classified into three groups: Rhodophyta (red), Phaeophyceae (brown), and Chlorophyta (green). The global variety of all algae (micro and macro) is estimated to consist of over 164,000 species with roughly 9800 of them being seaweeds, just 0.17% of which have been domesticated for commercial exploitation [2]. In recent years, seaweed has gained in popularity, making it a more versatile food item that may be used directly or indirectly in preparation of dishes or beverages [3]. Many types of seaweed are edible, they provide the body with a different variety of vitamins and critical minerals (including iodine) when consumed as food, and some are also high in protein and polysaccharides [4].

Seaweeds are now used in several industrial products as raw materials such as agar, algin, and carrageenan, but they are still widely consumed as food in several nations [5]. Seaweeds are frequently subjected to harsh environmental conditions with no visible damage; as a result, the seaweed generates a wide variety of metabolites (xanthophylls, tocopherols, and polysaccharides) to defend itself from biotic and abiotic factors such as herbivory or mechanical aggression at sea [6]. Please note that the content and diversity of seaweed metabolites are influenced by abiotic and biotic factors such as species, life stage, nutrient enrichment, reproductive status, light intensity exposure, salinity, phylogenetic diversity, herbivory intensity, and time of collection; thus, fully exploiting algal diversity and complexity necessitates knowledge of environmental impacts as well as a thorough understanding of biological and biochemical variability [7,8].

Seaweeds and their products are particularly low in calories but high in vitamins A, B, B2, and C, minerals, and chelated micro-minerals (selenium, chromium, nickel, and arsenic), as well as polyunsaturated fatty acids, bioactive metabolites, and amino acids [9]. Although current research revealed that the amount of specific secondary metabolites dictates the effective bioactive potential of seaweeds, phenolic molecules are prevalent among these secondary metabolites [10]. Furthermore, integrating seaweed into one’s daily diet has been linked to a lower risk of a range of disorders, including digestive health and chronic diseases such as diabetes, cancer, or cardiovascular disease, according to research mentioned by [11]. As a result, incorporating seaweed components into the production of novel natural drugs is one of the goals of marine pharmaceuticals, a new discipline of pharmacology that has evolved in recent decades.

The $4.7 billion worldwide algae products market is predicted to increase at a compound yearly growth rate of 6.3% to $6.4 billion by 2026. North America has the highest proportion of the algae market [6]. Functional and nutritional attributes, as well as the potential sustainability benefits of algae, are driving demand and positioning it as a promising food of the future. The potential uses of different algae are numerous: generation of energy [12], the biodegradation of urban, industrial and agricultural wastewaters [13,14], the production of biofuels [15], the exclusion of carbon dioxide from gaseous emissions via algae biofixation [16], the manufacturing of ethanol or methane, animal feeds [17], raw material for thermal treatment [18], organic fertilizer or biofertilizer in farming [17]. The high protein content and health advantages have fueled an interest in foods derived from entire algae biomass [19]. Algae can be used as functional ingredients to boost food’s nutritional value [20]. In cosmeceuticals, marine algae have received a lot of interest [21]. Seaweeds are one of the most abundant and harmless marine resources, with little cytotoxicity effects on people. Marine algae are high in bioactive compounds, which have been demonstrated to have significant skin advantages, especially in the treatment of rashes, pigmentation, aging, and cancer [22]. The use of algal bioactive components in cosmeceuticals is growing quickly since they contain natural extracts that are deemed harmless, resulting in fewer adverse effects on humans. Marine algae were used as a medicine in ancient times to treat skin problems such as atopic dermatitis and matrix metalloproteinase (MMP)-related sickness [22]. In summary, marine algae represent a promising resource for cosmeceutical production.

This review aimed to study the bioactive compounds in seaweeds and the role of these compounds as antioxidants, anti-inflammatory, anti-cancer, antimicrobial and anti-diabetic activities.

## 2. Seaweed Resources

The word “seaweed” has no taxonomic importance; rather, it is a popular term for the common large marine algae.

### 2.1. Brown Seaweeds

Phaeophyceae have not been well investigated, despite the fact that they have been shown to offer several health benefits. Fucoxanthin (Fuco), the principal marine carotenoid (Car), is a commercially important component of brown seaweeds, in addition to sodium alginate. Fuco contains anti-inflammatory properties. The presence of the xanthophyll pigment fucoxanthin, which is higher than chlorophyll-a, chlorophyll-c, -carotene, and other xanthophylls, gives these seaweeds their brown color [23]. Because of its bigger size and ease of collecting, brown seaweed is used in animal feed more often than other algae species. Brown algae are the largest seaweeds, with some species reaching up to 35–45 m in length and a wide range of shapes. *Ascophyllum*, *Laminaria*, *Saccharina*, *Macrocystis*, *Nereocystis*, and *Sargassum* are the most prevalent genera. Sargassum as a member of brown seaweeds is low in protein, but high in carbs and easily accessible minerals. They are high in beta-carotene and vitamins, and they are free of anti-nutrients [24].

### 2.2. Red Seaweeds

These algae are red because of the pigments phycoerythrin and phycocyanin. The walls are made of carrageenan and cellulose agar. Both of these polysaccharides with a lengthy chain are commonly employed in the industry. Coralline algae, which secrete calcium carbonate on the surface of their cells, are an important category of red algae. *Chondrus*, *Porphyra*, *Pyropia*, and *Palmaria* are some of the most common red algae genera. The antioxidant activity of Phaeophyta (brown seaweeds) is higher than that of green and red algae [25].

### 2.3. Green Seaweeds

The majority of the species are aquatic, living in both freshwater and marine habitats. The green color of these algae is due to chlorophyll-a or chlorophyll-b. Some of them are terrestrial, meaning they grow in soil, trees, or rocks. Ulva is one of the most common green seaweeds. *Ulva*, *Cladophora*, *Enteromorpha*, and *Chaetomorpha* are the most common genera. Green algae thrive in regions with lots of light, including shallow waterways and tide pools. *Ulva* sp. has a high protein content (typically > 15%) and a low energy content and is abundant in both soluble and insoluble dietary fiber (glucans) [26]. The main types of seaweeds are shown in Figure 1.

## 3. Bioactive Compounds

The chemical composition of algae varies depending on the species, cultivation location, meteorological conditions, and harvesting period. Because of the broad diversity of compounds produced by seaweeds, they are currently considered to be prospective organisms for contributing new physiologically active chemicals for the production of novel food (nutraceutical), cosmetic (cosmeceutical), and medical compounds. Polyphenolic compounds, carotenoids, minerals, vitamins, phlorotannins, peptides, tocotrienols, proteins, tocopherols, and carbohydrates (polysaccharides) are considered to be a great variety of bioactive compounds (Figure 2).

### 3.1. Polysaccharides

Seaweeds have a significant carbohydrate component in their cell membranes, or these polysaccharides are unique to every variety from algae: Brown alginate contains fucoidan; green Ulvan or red agar contains carrageenan. Polysaccharides are becoming increasingly popular as a result of their physicochemical properties [27]. Polysaccharides are biopolymers created from natural resources that have developed as a sustainable and environmentally friendly alternative to typical polymers and plastics. They are also known as an energy store and structural molecules in a variety of species, including plants and marine organisms. Polysaccharides are the major macromolecule in seaweed, accounting for more than 80% of its weight. Polysaccharides are classified into two types based on where they are found in seaweeds: cell-membrane polysaccharides or storage polysaccharides. With the exception of accumulating carbohydrates found in cell plastids, the majority of seaweed polysaccharides are cell-membrane polysaccharides. At present, they can be classed as food-grade or non-food-grade polysaccharides, depending on how they are used [28].

#### 3.1.1. Role of Polysaccharides in Medicine

Algal polysaccharides differ from those found in terrestrial plants because they include unique poly-uronides, some of which are pyruvylated, methylated, sulfated, or acetylated. Sulfated polysaccharides including fucan sulfate, ulvan, and carrageenan have received the most interest because of their biological features [29]. Some of polysaccharide’s structures are presented in Figure 3. Sulfated polysaccharides (SPS) are found in edible seaweeds such as ulvan (Chlorophyta), fucoidan (Phaeophyta), or carrageenan (Rhodophyta), and have numerous applications in pharmaceutical, nutraceutical, and cosmeceutical sectors. Antioxidant, anticancer, anti-inflammatory, anti-diabetic, anticoagulant, immunomodulatory, or anti-HIV activities have been discovered in SPS. The interaction between polysaccharide or intestinal microbes is widely credited with these actions, indicating functional or therapeutic feature of sulfated polysaccharides [30]. In most circumstances, smaller molecular weight SPS has more antioxidant activity than high molecular weight SPS because proton donor action occurs in cells in low molecular weight SPS. Furthermore, this antioxidant property is vital in preventing generation of free radicals in cell, which prevents oxidative cell wall damage [31]. The antigenotoxic property of alginate oligosaccharide in form of nanocomposites extracted from brown alga has received significant attention [32]. Table 1 shows some of the activities and qualities of polysaccharides from seaweeds that are useful as antioxidants and anticancer agents.

Carrageenans are polysaccharides present in cell walls of red algae that are classified into three categories based on their sulfation level: iota, kappa, or lambda [41]. Carrageenans, galactan, or xylomannan sulfates discovered in red seaweeds have antimicrobial effects that prevent viruses from interacting with cells by inhibiting the formation of structurally similar complexes [42]. Carrageenans derived from Hypnea spp. (including green alga *Ulva lactuca*) have antioxidant and antiviral characteristics, as well as strong hypocholesterolemic capabilities, by lowering cholesterol or sodium uptake whereas raising potassium absorption [43]. Agar is polysaccharide made up of agaropectin or agarose, which are both derived from red seaweeds and have structural or functional characteristics that are comparable to carrageenans [41]. Porphyran, a polysaccharide produced from red Porphyra spp., was shown to have anticancer, immunoregulatory, and antioxidant effects [44].

Sulfated polysaccharides such as galactose, glucose, rhamnose, glucuronic acid, or arabinose isolated from the microalgae *Spirulina platensis*, as well as those speculated from red algae *Gracilariopsis lemaneiformis* (i.e., 3,6-anhydro-l-galactose or d-galactose) demonstrated antiviral and antitumor action [44]. Fucoidan polysaccharides, usually manufactured by brown algae, such as *Ascophyllum nodosum*, *Laminaria japonica*, *Viz fucusvesiculosus*, *Fucus evanescens*, *Sargassum thunbergi*, or *Laminaria cichorioides*, were shown to reduce blood cholesterol levels and deter metabolic syndrome [43]. Antiproliferative, antiviral, anti-peptic, antioxidant, anticanceranti-coagulant, antithrombotic, anti-inflammatory, or antiadhesive characteristics are all found in algae fucoidans. They also have potent anticancer properties or can prevent lung cancer metastasis through hindering matrix metalloproteinases (MMPs) or Vascular Endothelial Growth Factor (VEGF) [45]. Fucoidans may have a synergistic impact on currently used anticancer drugs [46]. To improve the efficacy of existing conventional treatments, these polysaccharides can be added into or mixed with them. *Caulerpa lentilifera*, *Eucheuma cottonii*, *Ahnfeltiopsis concinna*, *Chondrus ocellatus*, *Sargassum polycystum*, *Ulva fasciata*, *Gayralia oxysperma*, or *Sargassum obtusifolium* soluble dietary fibers have been found to prevent metabolic syndrome or lower blood cholesterol levels [43].

Alginate (β-d-mannuronic acid, α-l-guluronic acid, d-guluronic, or d-mannuronic) is non-sulfated polysaccharide isolated from dark brown seaweed *Laminaria digitata* that is commercially accessible (in acid and salt forms) [41]. Alginates isolated from brown have a nutritional function or are beneficial to gut health, donating to water binding, fecal bulking, or reduction in colon transfer time that is an important indicator through colon cancer prevention, according to previous studies [47]. Furthermore, because of their binding nature, alginates alter mineral bioabsorption, aid in maintaining body weight, discourage overweight and obesity, and lower hypertension [41].

#### 3.1.2. Role of Polysaccharides in Food Industry

While the global market for healthy ingredients expands, there is significant interest in the identification of new functional food ingredients from various natural sources [48]. As a result, the prospect of employing algae-derived molecules to create novel functional food products has piqued the interest of many people in recent years. The largest and most often used hydrocolloids from marine algae in the food industry include agars, alginates, and carrageenans, as illustrated in Table 2.

Agar is a type of phycocolloid formed of agarose (a linear polysaccharide) and a heterogeneous combination of smaller molecules (agaropectin). Agar is a widely recommended food additive in the USA and in Europe (E406), and cannot be digested into the gastrointestinal system in humans due to the lack of α/β-agarases [57]. Furthermore, gut bacteria can convert it to d-galactose [58]. At low doses, agar is an excellent gelling agent, capable of forming a brittle, stiff, and thermally reversible gel [59].

Surprisingly, agarose is the primary gelling agent in agar. In this manner, hydrogen bonding between nearby D-galactose and 3,6-anhydro-L-galactose create agar gel along its linear chains of agarose with repeating units. The food sector uses 90% of the agar produced for its gellifying characteristics. It is used as a gelling agent in the culinary, food, and confectionery sectors to produce Asian traditional foods, canned meats, confectionery jellies, and aerated items such as marshmallows, nougat, and toffees [60]. Agar is commonly used as a food additive in the production of dishes that require warming before consumption, such as cake, sausage, roast pig, and bacon [61]. Agar fluid gels can be used to make foams with excellent stability to replace fat in whipped desserts [61].

Alginates, such as agar, are commonly used in the food manufactures for gelling, thickening, stabilizing, and film formation. In contrast to other hydrocolloids, alginates are unique in their cold solubility, allowing the creation of heat/temperature-independent non-melting gels, cold-setting gels, and freeze–thaw-stable gels [62].

Carrageenan is commonly used in dairy products such as cheese and chocolate milk to provide thickening, gelling, stabilizing, and strong protein-binding characteristics [63]. Carrageenan was used in dairy products at low doses due to its exceptional ability to link milk proteins. This hydrocolloid was capable of keeping milk solids suspended and therefore stabilize them. The meat industry is another area where carrageenan (mostly manufactured by Eu-cheuma) is used. It is commonly used in the manufacture of hamburgers, ham, seafood, and poultry preparations, due to its water retention properties. Carrageenan is also found in aqueous gels such jelly, fruit gels, juices, and marmalade [61]. Carrageenans, as cryoprotecting agents, play an important role in the structural and textural stability of frozen foods. Additionally, k-carrageenan was used as a supplementary stabilizer in an ice cream mix [64].

#### 3.1.3. Role of Polysaccharides in Cosmeceuticals

In algal tissues, there are numerous forms of bioactive polysaccharides. These chemicals are often moisturizing and antioxidant substances that are employed in cosmeceuticals as shown in Table 2. They are also commonly employed in emulsions as gelling agents and stabilizers [65]. Agar is a common ingredient in creams, used as an emulsifier and stabilizer, and to control the moisture content in cosmetic products such as hand lotions, deodorants, foundations, exfoliant/scrub, cleansers, shaving creams, anti-aging treatments, facial moisturizer/lotions, liquid soaps, acne treatments, body washes, and face powder [66]. Alginates are commonly used as gelling agents in drugs and cosmetics, as thickeners, protective colloids, or emulsion stabilizers, and are effective for hand gels and lotions, ointment bases, pomades and other hair products, toothpastes, and other products due to their chelating characteristics. Alginates can also be used to make a skin-protecting barrier lotion to avoid dermatitis. This type of cream produces flexible films with increased skin adhesion and is an appropriate component in beauty masks or facial packs [67,68].

Carrageenans are derived from several carrageenophytes, including *Betaphycus gelatinum*, *Chondrus crispus*, *Eucheuma denticulatum*, *Gigartina skottsbergii*, *Kappaphycus alvarezii*, *Hypnea musciformis*, *Mastocarpus stellatus*, *Mazzaella laminaroides*, *Sarcothalia crispata*, from the order Gigartinales (Rhodophyta). This phycocolloid is found in dentifrices, lotions, hair products, lotions, medications, sunscreens, shaving creams, shampoos, deodorants sticks, sprays, and foams. Over 20% of carrageenan manufacture is used in the pharmaceutical and cosmetic industries [69].

The usage of laminarin in cosmetics is based on its bioactive qualities rather than its physical characteristics. In terms of use, laminarin is commonly found in anticellulite cosmetics [70]. Fucoidan can be effectively “cooked” out of edible seaweed by heating it in water for 20–40 min. It appears to lower the strength of the inflammatory process and facilitate speedier tissue repair after injuring or surgical trauma when ingested. As a result, it is recommended for muscle and joint injuries (such as sports injuries), falls, bruises, deep wounds, and surgery [71]. These sulfated polysaccharides are gaining popularity due to their numerous bioactivities, which include anticoagulant, antithrombotic, anti-inflammatory, skin protection against ultraviolet radiation, tyrosinase receptor, anticancer, antimicrobial, anti-obesity, antidiabetic, antioxidative, and antihyperlipidemic properties [72,73].

According to an ulvans patent, rhamnose and fucose have synergistic skin protecting and therapeutic benefits against skin aging [74]. The technique of ulvan gel production is complex, involving the development of spherically shaped ulvan molecules in the presence of boric acid and calcium ions [75]. Ulvans have moisturizing, protecting, anticancer, and antioxidative effects in addition to their ability to form gels [76]. The chemical and physicochemical features of ulvan make it an appealing choice for innovative functional and biologically useful polymers in the pharmaceutical, cosmeceutical, agriculture, and food industries [75].

### 3.2. Protein and Amino Acids

Protein content in seaweed varies by species, season, and geographic location, and can be as high as 45% DW. The contents of peptides, proteins, or amino acids in seaweed are affected by seasonal fluctuations and habitat; in general, red algae have larger concentrations (up to 47%) than green algae (around 9 and 26%), while brown algae have low amounts (3–15%) [77]. The difference in the amounts of proteinas and amino acids in some seaweeds are illustrated in Table 3 and Table 4. All essential and non-essential amino acids are found in the proteins of the three macroalgae groups [78]. Seaweed protein and bioactive peptides have a variety of health benefits as well as significant antioxidant activity, especially through compounds with low molecular weight compounds that are far secure than produced substances or have less adverse impacts [79,80].

Various seaweeds contain amino acids such as valine, leucine, isoleucine, or taurine which have potential biological action as antioxidants [92,93]. Acidic amino acids aspartic acid or glutamic acid is abundant in most seaweed species, and they comprise most essential amino acids [94]. While algal proteins were being thought to consist of threonine, tryptophan, sulfur amino acids (cysteine and methionine), lysine, or histidine-limiting amino acids, their overall levels are larger than in terrestrial plants [95]. Furthermore, amino acids are required for the production of hormones and nitrogenous low molecular weight compounds, both of which are important biologically. Amino acids can be used to help treat some disorders since they have distinct physiological roles. Supplementing with methionine, for example, can help people with multiple sclerosis [96]. Despite the fact that seaweed proteins contain low amounts of some essential amino acids, these seaweeds could be introduced to cereal foods such as pasta to enhance the amino acid composition [97].

Macroalgal species such as *Chlorella* sp., *Dunaliella tertiolecta*, *Aphanizomenon flosaquae* and *Spirulina plantensis*, due to their high protein content or nutritive quality, are often used as human food sources [98]. Endogenous (threonine, serine, aspartic acid, proline, glutamic acid, or glycine) and exogenous (histidine, lysine, isoleucine, methionine, phenylalanine, leucine, valine or threonine) amino acids are abundant in some algae species [43]. Ulva spp. has glutamic or aspartic acid (26–32% amino acid), *Ulva australis* has taurine or histidine, *Himanthalia elongata* (sea spaghetti) *Palmaria palmata* (Dulse) and have a lot of glutamic acid, serin or alanine, and *Sargassum vulgare* has lot of methionine [99]. Several applications of seaweeds protein are illustrated in Table 5.

#### 3.2.1. Role of Proteins and Amino Acid in Medicine

Furthermore, mycosporine-like amino acids (MAAs) were revealed in a variety of species, most notably Rhodophyta: *Chondrus crispus* spp., *Grateloupia lanceola*, *Porphyra*/*Pyropia* spp., *Solieria chordalis*, *Asparagopsis armata*, *Palmaria palmata*, *Gracilaria cornea*, Gelidium, or *Curdiea racovit* [106,107,108]. Phycobiliproteins are made up of phycobilins, which are proteins that are covalently attached to chromophores [43]. Such water-soluble proteins have antioxidant properties and could be used as a natural food colorant [26]. PC, blue-colored phycobiliprotein derived mostly from cyanobacteria Arthrospira spp., or PE (pink-colored protein pigment) derived from cyanobacteria Lyngbya spp., both demonstrated anticancer activity upon A549 lung cancer cells [22]. Glycoproteins were also proteins found in marine algae which are made up from proteins linked to carbohydrates. Rhamnose, galactose, glucose, and mannose make up 36.24% of glycoproteins, with a mole ratio of 38:30:26:6 [109].

Protein concentrations are high in *Rhizoclonium riparium*, *Dictyota caylinica*, *Enteromorpha intestinalis*, *Catenella repens*, *Gelidiella acerosa*, *Polysiphonia mollis*, *Capsosiphon fulvescens*, *Osmundea pinnatifida*, *Sphaerococcus coronopifolius*, *Ulva lactuca*, *Gelidium microdon*, *Fucus spiralis*, *Pterocladium capillacea*, or *Ulva compressa* [110]. Anti-aging, antioxidant, anti-tumor, anti-inflammatory or protective qualities of proteins make them valuable in the prevention and treatments of neurological illnesses, DNA replication, gastric ulcers, improve response, molecule transfer, or biochemical reaction catalysis [45]. According to Cicero et al. [111], bioactive peptides can increase biological defenses against oxidative stress and inflammatory illnesses, hence boosting the real frame of nutraceutical and functional meals. As a result, MAAs have wide range of properties, such as ability to act like natural sunscreens, anti-inflammatory, antioxidants or anti-aging agents, skin renewal stimulators, cell proliferation activators, and so on, making it attractive or secure option for cosmetic industries or pharmaceutical [112].

#### 3.2.2. Role of Proteins and Amino Acid in Cosmeceuticals

Because several amino acids are components of the natural moisturizing factor (NMF) in human skin, they are commonly used as moisturizing agents in cosmetic preparations [113,114]. MAA content is higher in the summer and at a mild depth (0–1 m). MAAs have the ability to be used in cosmetic products and uses as ultraviolet protectors and cell proliferation stimulators [115].

Algae protein concentration differs significantly among the different algae groups (brown, red, and green). Brown algae have a lower protein concentration (5–24%) of dry weight, while red and green algae have a greater protein concentration (10–47%) of dry weight [116]. Holdt and Kraan [107] show that protein, peptide, and amino acid concentration, like other bioactive components of algae, is affected by a variety of circumstances, including seasonal change. During the months from February to May, for example, brown algae Saccharina and Laminaria had the highest protein content [107]. A similar trend was observed in red algae species, with a high concentration of protein in the summer and a significant decrease in the winter [116]. Algae proteins are high in glycine, arginine, alanine, and glutamic acid, and they include essential amino acids at amounts comparable to FAO/WHO needs. Lysine and cystine are their limiting amino acids [117]. Taurine, laminin, kainoids, kainic and domoic acids, and several mycosporin-type amino acids are also found in algae [118]. Taurine is involved in several physiological activities in the human body, including immunomodulation, membrane stabilization, ocular development, and nervous system function [119]. Furthermore, kainic and domoic acids play a role in the control of neurophysiological functions [120].

### 3.3. Fatty Acids

Fatty acids (FAs) are required for all organisms to function normally. FAs are components of plasma membranes that serve as energy storage materials as well as signal molecules that control cell development and differentiation as well as gene expression. Elongation and desaturation can change the structure of FAs [121,122]. The quantity of unsaturated bonds in FA molecules determines their biological effects. Additionally, lipids are essential to transport and absorb fat-soluble vitamins (i.e., A, E, D or K). PUFAs (25–60% of total lipids), glycolipids, phytosterols, phospholipids, or fat-soluble vitamins are all found in low concentrations (1–5% of dry weight) in seaweed lipids (vitamin A, D, E or K, carotenoids) [1]. Several seaweeds have a greater total lipid concentration above 10% of dry weight; however, 50% of these lipids are in the form of extractable fatty acids in the brown alga *Spatoglossum macrodontum*. In addition, *S. macrodontum* showed the maximum fatty acid concentration (57.40 mg g^−1^ DW) and a fatty acid profile rich in saturated fatty acids with a higher concentration of C18:1, making it an excellent biofuel feedstock. Similarly, the green seaweed *Derbesia tenuissima* possesses significant quantities of fatty acids (39.58 mg g1 DW), but with a greater amount of PUFA (n-3) (31% lipid) that can be used as nutraceuticals or fish oil substitutes [123]. The lipid algae concentration is low (1–5%), with neutral lipids and glycolipids dominating. Because algae generate long-chain polyunsaturated fatty acids, including eicosapentaenoic acid (EPA) and docosahexaenoic acid (DHA), the amount of essential fatty acids in algae is greater than in terrestrial plants [124]. In general, red algae have higher concentrations of EPA, palmitic acid, oleic acid, and arachidonic acid than brown algae, which have greater amounts of oleic acid, linoleic acid, and α-linolenic acid but lower amounts of EPA. Green algae have more linoleic acid and α-linolenic acid, as well as palmitic, oleic, and DHA [125]. Both red and brown algae contain omega-3 and omega-6 fatty acids [126]. The different in the amounts of lipid in different seaweeds are illustrated in Table 6.

#### 3.3.1. Role of Fatty Acids in Medicine

There is an increasing need to assess new food sources that do not involve overexploitation of terrestrial ecosystems [133]. Seaweeds have a lipid output of 0.61% to 4.15% dry weight (DW) on average. Some seaweed species, on the other hand, can have greater levels since they are a strong source of unsaturated fatty acids. Although seaweed has lower lipid content than marine fish, their abundance in coastal areas makes it a viable source of functional lipid. Recent studies indicated that the levels of total lipid (TL) or omega-polyunsaturated fatty acids in seaweeds vary seasonally, reaching up to 15% TL per DW or more than 40% omega-3 PUFAs per total fatty acids [134]. Brown seaweed lipids, on the other hand, contain up to 5% fucoxanthin. Anti-obesity activities of fucoxanthin have been demonstrated. It also reduces insulin resistance and lowers blood glucose levels significantly. Brown seaweed lipids are found in brown seaweed, according to a study. Excess fat builds up in abdomen white adipose tissue (WAT) is dramatically decreased, or glucose levels are regained to average limits in obesity/diabetes model mice due to presence of fucoxanthin in lipids [135].

On the other hand, the group of lipid bioactive chemicals known as sterols is another appealing lipid bioactive substance found in marine sources. Sterols extracted from macro- or microalgae, as well as other marine invertebrates, were researched extensively by [136]. Previously, it was discovered that sterols and several of their derivatives have a key role in decreasing low-density lipoproteins (LDL) cholesterol levels in vivo. Anti-inflammatory and antiaterogenic action are two further bioactivities linked to sterols. Phytosterols (C28 and C29 sterols) are also key precursors of a wide range of chemicals, including vitamins. Ergosterol, for example, is a precursor to vitamin D2 and cortisone [137].

Omega-3 (eicosapentaenoic acid, docosahexanoic acid, stearidonic acid, -linolenic acid) and omega-6 (arachidonic acid, -linoleic acid, -linoleic acid) are the most common polyunsaturated fatty acids (PUFAs) [1]. Essential fatty acids (EFAs) are nutraceuticals that are combined with nutritional supplements or used as part of healthy food [41]. Food and Drug Administration (FDA) declared in 2004 which foods including PUFA omega-3 substances are medicinally essential, as they provide therapeutic properties byregulating blood pressure, membrane fluidity, or blood clotting; (ii) lowering risk of cardiovascular disease, osteoporosis, or diabetes; (iii) correcting brain or nervous system development and function [138]. Marine algae were found to have elevated high levels from PUFAs (α-linolenic acid, γ-linoleic, α-linoleic acid, stearidonic acid, arachidonic acid, and icosapentaenoic acid) [1]. Moreover, a previous study asserted that green seaweeds such as *Ulva pertusa* possess a high concentration of hexadecatetraenoic, oleic, and palmitic acids [139]. Additionally, *Undaria pinnatifida* contains significant levels of eicosapentaenoic acid, docosahexanoic acid, and monounsaturated fatty acids (C12:1 (lauroleic acid), C14:1 (myristoleic acid), C16:1 (palmitoleic acid), C17:1 (cis-10-heptadecenoic acid), and C18:1 (cis-10-hepta (oleic acid) [140].

Upwards of 200 phytosterols (662–2320 mg/g dry weight) were discovered through marine algae. Phytosterol derivatives are abundant in brown algae such as *Laminaria japonica*, *Agarum cribosum*, or *Undaria pinnatifida* (for example, fucosterol, which accounts for 83–97 percent of total phytosterol content) [141,142]. Phospholipids in seaweed contain about 10–20% total lipids which seem to be more resistant to oxidation and contain elevated concentration from FAs such as eicosapentaenoic or docosahexanoic acid [43]. Glycolipids make up more than half of all algal material and are characterized by high levels of n-3 PUFAs (e.g., monogalactosyldiacylglycerides, digalactosyldiacylglycerides or sulfoquinovosyldiacylglycerides) [41]. Carotenoids are a group of lipophilic colorful chemicals found in nature that include lutein, lycopene, canthaxanthin, β-carotene, or astaxanthin [143]. Furthermore, these properties make algal lipids more bioavailable or provide a variety of health benefits to people or animals [109].

#### 3.3.2. Role of Fatty Acids in Foods

Microalgae have a high PUFA content. They are fatty acids with many double bonds in the carbon chain and have numerous useful qualities. Microalgae may produce members of the PUFAs ω-6 family, such as linoleic acid (LA), γ-linolenic acid (GLA), and arachidonic acid (ARA), as well as members of the PUFAs ω-6 family, such as α-linolenic acid (ALA), eicosapentaenoic acid (EPA), and docosahexaenoic acid (DHA) [144,145]. Many microalgae manufacture the long chain of -3 PUFAs, with yields exceeding 20% of total lipids. The microalgae most commonly employed for the formation of algal oil rich in ω-3 and biomass are marine members of the Thraustochytriacea and Crythecodiniacea families [146].

Because of their obvious benefits to tissue integrity and health, they are vital ingredients for food additives and feeds. Microalgae such as *Chlorella vulgaris*, *Arthrospira platensis*, *Haematococcus pluvialis*, and *Dunaliella salina* have been identified as safe or permitted as human and animal food additives. *Scenedesmus almeriensis* and *Nannocholoropsis* sp. are two more species that have been investigated but have not yet been commercialized [147].

*Crypthecodinium*, *Schizochytrium*, *Thraustochytrids*, and *Ulkenia* microalgal species are employed in the manufacture important fatty acids [148]. DHA-rich oil derived from *Crypthecodinium cohnii* is commercially accessible and contains 40–50% DHA with no EPA or other longchain PUFAs [149]. Schizochytrium species that synthesize DHA and EPA are currently employed as an adult dietary supplement in food and drinks, health foods, animal feeds, and foodstuffs products such as cheeses, yogurts, spreads and sauces, and breakfast cereals. This microalga’s essential fatty acids are used as supplements in diets for pregnant and nursing women, as well as cardiovascular patients [149].

#### 3.3.3. Role of Fatty Acids in Cosmeceuticals

Algae fatty acids and other lipophilic chemicals are also anti-allergic, antioxidant, and anti-inflammatory [150]. Furthermore, lipids can act as moisturizing ingredients substances, protecting the skin from water loss [151]. Many fatty acids, including lauric acid, myristic acid, palmitic acid, and stearic acid, can be used as raw materials. Furthermore, FAs are skin components that play a crucial role in the maintenance of skin integrity [152].

Waxes are classified as fatty esters, which are a type of fatty acyl [153]. *Euglena gracilis* is a microalga that produces a large quantity of wax-ester as a byproduct of the biodegradation of storage polysaccharides. These wax-esters are now used in biofuel generation but could possibly be useful in cosmetics [154]. Waxes, for example, are important components in lipsticks because they give the stick sufficient rigidity, hardness, stability, and texture. Today’s lipsticks can be made with a range of waxes. Alkenones are a class of lipids, long-chain ketones that are produced by haptophyte microalgae such as *Isochrysis* sp. and employed as structuring agents in some cosmetic preparations in place of animal-derived and petroleum-derived waxes. They are a vegan and recyclable marine-based component that will meet customer demands. Because alkenones can be made in a variety of locales, their supply is not as limited as that of some other waxes. Given their waxy structure and relatively high melting point, alkenones may offer an appealing class of natural chemicals with potential applications in a wide range of cosmetic and skin care products [155]. Table 7 highlights the applications of lipids.

### 3.4. Pigments

Natural pigments are necessary for photosynthesizing algal metabolism, or macroalgae are divided into three groups depending on pigment content: Phaeophyceae (brown algae), Chlorophyceae (green algae), or Rhodophyceae (red algae) are three families of algae (red algae) [139]. Macroalgae can produce three fundamental types of organic pigments: chlorophylls, carotenoids, or phycobilins [140]. Macroalgae that are wealthy in chlorophylls a or b seem green, whereas algae appear greenish-brown owing to a combination of fucoxanthin (carotenoid), and algae appear red owing to combination of chlorophylls a, c, or d, and phycobilins. Chlorophylls are natural lipid-soluble greenish pigments with porphyrin ring [139]. The chemical structures of different types of pigments in seaweeds are presented in Figure 4.

Carotenoids have received much interest and are used in nutritional supplements, fortified foods, animal feed, pharmaceuticals, or cosmetics because of their antioxidant and antimicrobial characteristics, which assist to decrease the prevalence of cardiovascular diseases, ophthalmologic diseases, or cancer [138]. Carotenoids are lipophilic, linear polyenes in two categories: (i) carotenoids, carotenoids, and lycopene; (ii) xanthophylls (e.g., antheraxanthin, zeaxanthin, lutein, fucoxanthin, violaxanthin) [167]. *Ascophyllum nodosum*, *Cladosiphon okamuranus*, *Fucus serratus*, *Chaetoseros* sp., *Ishige okamurae*, *Ecklonia stolonifera*, *Himanthalia elongata*, and *Fucus vesiculosus* all contain carotenoid. It is more efficient upon Gram-positive bacteria (like, *Streptococcus agalactiae*, *Staphylococcus aureus*, *Proteus mirabilis*, *Pseudomonas aeruginosa*, *Staphylococcus epidermidis*, or *Serratia marcescens*) and Gram-negative bacteria (like, *Klebsiella pneumoniae*, *Klebsiella oxytoca*, *Serratia marcescens*, *Acinetobacter lwoffii*, *Pseudomonas aeruginosa* or *Escherichia coli*) [139].

Phycobiliproteins are naturally fluorescent, water-soluble proteins classified as PC (blue pigment), PE (red pigment), and allophycocyanins (light-blue pigment), with PE being most common in several red macroalgae species [139]. Algae rich in phycobiliproteins include *Spirulina*, *Botryococcus*, *Chlorella* and *Nostoc*. These pigments were discovered to have anti-obesity, anti-inflammatory, anti-angiogenic, antioxidant, anti-carcinogenic or neuroprotective activities in a recent study [168]. Table 8 illustrates the role of different carotenoids in human health.

### 3.5. Phenolic Compounds

Phenolic acids, tannins, flavonoids, and catechins are some of the phenolic compounds found in marine algae. The method of phenolic chemical extraction and the yield are strongly dependent on seaweed species. Brown seaweeds (Pheophyceae: P) are known for their high content of phlorotannins, complicated polymers made up of oligomers of phloroglucinol (1,3,5-trihydroxybenzene), while red or green seaweeds (Rhodophyceae: R) are known for their phenolic acids, flavonoids or bromophenols [10]. Polyphenols extracted from seaweeds were linked to variety of biological functions, containing antimicrobial, anticancer, antiviral, anti-obesity, antitumor, antiproliferative, antidiabetic, anti-inflammatory, or antioxidant effects [10]. Previous studies [101,188] demonstrated the anti-inflammatory activity of polyphenol-rich fraction derived from Rhodophyceae. Furthermore, phlorotannins and bromophenols derived from green or red algae possess strong inhibitory activity upon in vitro cancer cell proliferation or in vivo tumor growth, as well as antidiabetic and antithrombotic activities in vitro.

The phenolic active ingredients in seaweeds differ depending on whether they are red, green, or brown. Different phyla create different chemicals; for example, brown seaweeds produce phlorotannins, but red seaweeds produce a greater range of mycosporine-like amino acids (MAAs) than green species [189,190]. As a result of cellular mechanisms and genetic codification, the synthesis and diversity of phenolic chemicals are intimately tied to the seaweed taxonomic group and individual species [191]. Furthermore, phenolic acids such as benzoic acid, p-hydroxybenzoic acid, salicylic acid, gentisic acid, protocatechuic acid, vanillic acid, gallic acid, and syringic acid have been found in the genus Gracilaria (Rhodophyta, red alga) [192,193]. Phlorotannins are well-known phenolic chemicals that brown seaweeds produce [194]. Flavonoids such as rutin, quercitin, and hesperidin were detected in many Chlorophyta, Rhodophyta, and Phaeophyceae species [195]. Chondrus crispus and *Porphyra*/*Pyropia* spp. (Rhodophyta), as well as *Sargassum muticum* and *Sargassum vulgare* (Phaeophyceae), may synthesis isoflavones, as can daidzein and genistein [196]. Furthermore, several flavonoid glycosides were found in the brown seaweeds *Durvillaea antarctica*, *Lessonia spicata*, and *Macrocystis pyrifera* (also known as *Macrocystis integrifolia*) [195].

Terpenoids are belonging to secondary metabolites discovered in seaweeds [190]. Meroditerpenoids (such as plastoquinones, chromanols, and chromenes) were discovered in brown seaweeds, primarily from the Sargassaceae family (Phaeophyceae). These compounds are produced in part from terpenoids and are distinguished by the presence of a polyprenyl chain connected to a hydroquinone ring moiety [197]. In Rhodomelaceae, red seaweeds manufacture phenolic terpenoids such as diterpenes and sesquiterpenes. *Callophycus serratus*, for example, synthesizes a particular diterpene called bromophycolide [198]. Some studies revealed the existence of phenolic and flavonoids acids in marine algae as seen in Figure 5 and the chemical structure of phenolics also presented in Figure 6.

### 3.6. Minerals

Seaweeds comprise greater numbers of important minerals, such as macroelements (e.g., Na, Ca, P, Mg, K) and trace minerals (like, Fe, Zn, Mn, Cu) due to their marine environment [118]. Minerals and cell wall polysaccharides (such as agar, alginic acid, alginate, or cellulose) play critical roles in the formation of human tissues or the regulation of crucial reactions as cofactors of some enzymes as cofactors among some enzymes [107]. As a result, seaweeds are important source of minerals and, when consumed regularly, have been recognized as advantageous functional foods (i.e., food supplements) [98]. It is worth noting that brown algae have greater mineral content than red algae [118].

Furthermore, elements such as Fe or Cu are found in higher concentrations in seaweeds than in meats and spinach [43]. Seaweeds were identified to be a promising supplier of iodine, which occurs at different chemical components, or brown algae, which contains more than 1% moisture content; its buildup in seaweed tissues may be 30,000 times greater than its concentration in sea water [45]. Iodine, which comes in a variety of forms, is anti-goiter, anticancer, antioxidant agent or a key nutrient in metabolic control. However, excessive intake may result in some unfavorable effects [43].

Green seaweeds have a Na^+^/K^+^ ratio of 0.9 to 1, red seaweeds have a ratio of 0.1 to 1.8, and brown seaweeds have a ratio of 0.3 to 1.5. This ratio was found to be especially low in *Palmaria palmata* (0.1) and Laminaria spp. (0.3–0.4) from Spain [199]. Because the World Health Organization (WHO) recommends a Na^+^/K^+^ ratio close to one, consumption of food products with this proportion or lower should be examined for healthy cardiovascular purposes [199]. In contrast, using seaweeds as NaCl replacements in processed meals could be a useful technique for reducing overall Na+ consumption while boosting intake of K^+^ and other lacking components that would otherwise not be present in NaCl salted foods. In addition to Na^+^ and K^+^, Ca^2+^ and Mg^2+^ intake is linked to cardiovascular health. Indeed, it was proposed that enough Mg^2+^ intake may lower blood pressure by acting as a calcium antagonist on smooth muscle tone, inducing vasorelaxation [200].

Green seaweeds accumulate Mg^2+^ more than Ca^2+^, whereas brown seaweeds do the opposite. In turn, with the exception of *Phymatolithon calcareum*, which can accumulate exceptionally high concentrations of Ca^2+^ [201], red seaweeds generally have lower, but balanced, amounts of these two minerals compared to the two other macroalgae types. It should be noted that the Ca/Mg ratio is also important in terms of calcium absorption because a lack of magnesium can result in a buildup of calcium in soft tissues, resulting in the production of kidney stones and the formation of arthritis [202].

Finally, phosphorus (P) levels appear to be similar in the three macroalgae groups, with values ranging from 0.5 to 7 g/kg DW. Notably, Fe is prevalent in all three macroalgae types, while Chlorophyta has a greater rate than Rhodophyta and Phaeophyta. However, at low doses, some species from the chlorophyta phylum (e.g., *Alaria esculenta*, *Saccharina latissima*, and *Fucus* spp.) might also be proposed to be a good source of Fe, as accumulation in some cases can exceed 1 g/kg DW [203]. In turn, the maximum Mn concentrations were found in red seaweeds, specifically *Chondrus crispus*, *Palmaria palmata*, and *Gracilaria* spp. [204]. Dawczynski et al. [205] also described the preferential deposition of Mn by red macroalgae over brown macroalgae.

The production of seaweed-fortified foods with the goal of reducing NaCl consumption and increasing nutritive value has been notably emphasized in meat-based products. López-López et al. [206] conducted outstanding work in the reformulation of many meat products, partially replacing the application of sodium chloride with diverse species of edible seaweeds while retaining their textural and sensory qualities. This research group created meat emulsions, meat patties, and frankfurters enriched with *Undaria pinnatifida*, *Himanthalia elongata*, or *Porphyra umbilicalis* that were both low in Na+ and rich in K^+^, presenting Na^+^/K^+^ ratios below 1, which is much smaller than the ratios above 3 observed in their traditional recipes [207,208].

Furthermore, increasing the mineral content of meat, fish, and other animal-derived products can be accomplished by providing algae-supplemented diets to animals. Similarly, supplementing fish with seaweed-fortified meals has been shown to be an efficient way of increasing the iodine content of their fillets. Milk, dairy products, and, more recently, plant “milks” (e.g., soy, almond, oat, and rice) are another category of food products that play a critical role in the dietary routines of specific geographical areas of the world and, as such, are ideal candidates for macroalgae supplementation [209].

### 3.7. Vitamins

Vitamins are needed for a variety of skin functions and can be obtained from food or by topical application. Supplementation is indicated for skin protection against dryness and premature aging, aesthetic UV protection, and sebaceous gland secretory activity modulation. Vitamins are frequently found in skin care products or cosmetics. Vitamins A, C, E, K, or vitamin complex B seem to be the most essential or medically proven vitamins for skin photoaging treatment or prevention [77], as well as most abundant vitamins through algae have been vitamins A, B, C, or E [210].

Some seaweeds contain vitamins with several health benefits and antioxidant activity, which help to lower a variety of health issues such as high blood pressure, cardiovascular illnesses, and the risk of cancer [211]. Various seaweeds have been found to include water-soluble vitamins B1, B2, B12, and C, as well as fat-soluble vitamins E and β-carotene with vitamin A activity [212].

Vitamin A (β-carotene), in the form of retinol, has antioxidant and anti-wrinkle qualities [213] and is used in cosmetics to reduce hyperpigmentation or fine wrinkles on the face [214]. Vitamin B complex is found with higher concentrations in green or red seaweeds (B1, B2, B3 or niacine, B6, B9, B12, or folic acid) [215]. Active forms of vitamin B3 found in skincare products contain nicotinate esters, niacinamide, or nicotinic acid. Niacinamide is antioxidant that lowers hyperpigmentation (also caused by blue light) and enhances epidermal features by lowering trans-epidermal water loss [216]. Red algae or other species are good sources of vitamin B12, which has anti-aging characteristics or is required for hair, nail growth, or health in vegetarians [217].

Vitamin C is employed in cosmeceutical production because it contains L-ascorbic acid, the bioactive version of which is most well-known [213]. In this context, Ceramium rubrum and *Porphyr leucosticta* are red algae with elevated vitamin C content. This vitamin possesses antioxidant, antiviral, anti-inflammatory, antibacterial, detoxifying, or anti-stress properties when applied topically and could be used to improve tissue growth, repair blood vessels, teeth or bones [218]. A previous study found that if it is present in optimum concentration in cosmetic product, it can improve complexion, reduce pigmentation, and inflammation [219]. Vitamin C suppresses tyrosinase by interacting to copper ions that reduces melanogenesis, according to several studies [213].

Water-soluble vitamins, such as vitamin C, are abundant in *Ulva lactuca*, *Eucheuma cottonii*, *Caulerpa lentillifera*, *Sargassum polycstum*, and *Gracilaria* spp. and aid in the inhibition of low-density lipoprotein (LDL) oxidation and the creation of thrombosis/atherosclerosis [220]. Red algae have significantly higher levels of dried carotene (e.g., 197.9 mg/g in Codium fragile and 113.7 mg/g in *Gracilaria chilensis*) than other vegetables (e.g., 17.4 mg/g in *Macrocystis pyrifera*) [98], while brown seaweeds (e.g., *Undaria pinnatifida*) have greater concentrations of a-tocopherol/vitamin E (99% vitamins) than green and red seaweeds [107].

The primary fat-soluble vitamins (A and E) boost nitric oxide (NO) and nitric oxide synthase (NOS) activity, which aids in the prevention of CVDs [220]. Furthermore, vitamin E has antioxidant properties that can limit the oxidation of LDL [211]. Many disorders, such as chronic fatigue syndrome (CFS), anemia, and skin problems, are caused by a lack of water-soluble vitamins such as B12. Most terrestrial plants do not synthesize vitamin B_12_, but numerous prokaryotes that can synthesize vitamin B_12_ interact with seaweeds, and this interaction enhances vitamin levels in macroalgae [221]. Arthrospira (previously *Spirulina*) (Cyanobacteria) contains four times more vitamin B12 than raw liver [222]. Brown and green seaweeds are high in vitamin A, with 500–3000 mg/kg dry weight on average, but red algae have 100–800 mg/kg dry weight [223]. When compared to terrestrial plants, seaweeds such as *Crassiphycus changii* (previously *Gracilaria changii*), *Porphyra umbilicalis* (Rhodophyta), and *Himanthalia elongata* (Ochrophyta, Phaeophyceae) are high in vitamins [224]. Vitamins (A, B, C, D, and E) are found in seaweeds and are widely used in skincare [225].

Vitamin C minimizes the severity of allergic reactions to infection, boosts the immune system, regulates the creation of conjunctive tissue, and aids in the removal of free radicals. It also plays an important role in many diseases and disorders such as diabetes, atherosclerosis, cancer, and neurodegenerative problems [226]. The brown seaweeds Ascophyllum and *Fucus* sp. have higher levels of vitamin E (α-tocopherols) than other red and green seaweeds [227]. The seaweed *Macrocystis pyrifera* (Ochrophyta, Phaeophyceae) is high in vitamin E, similar to plant oils recognized for their vitamin E content, such as soybean oil (*Glycine max*), sunflower seed oil (*Helianthus annuus*), and palm oil (*Elaeis guineensis*) [227]. Vitamin E prevents the oxidation of low-density lipoprotein and is also effective in reducing the risk of cardiovascular disease [228].

## 4. Biological Activities

### 4.1. Antioxidant Activity

An imbalance in the creation and neutralization of free radicals causes oxidative stress, which leads to a variety of degenerative illnesses [229]. Several free radicals, particularly reactive oxygen species (ROS), were created in living organisms as a result of metabolic activity, and hence have an impact on health (Figure 7). ROS were formed in form of hydrogen peroxide (H_2_O_2_), superoxide radical (O_2_^−^), hydroxyl radical (·OH), or nitric oxide (NO). Oxidative stress causes unconscious or prominent enzyme activation, as well as oxidative damage for cellular systems [230]. ROS attack or damage important macromolecules including lipids membrane, proteins, or DNA, resulting in a variety of conditions include inflammatory or neurodegenerative diseases, diabetes mellitus, cancer, or severe tissue injuries [231,232] (Figure 7).

Antioxidants may have a favorable impact on human health because they may protect the body from damage caused via reactive oxygen species (ROS) [234]. To determine the antioxidant activity of marine derived bioactive peptides, researchers used electron spin resonance spectroscopy as well as intracellular free-radical scavenging assays.

ROS can produce several detrimental biological events, such as DNA oxidative lesions, membrane peroxidation, structural changes in proteins and functional carbohydrate, and so on. All of these structural and functional changes have direct clinical effects, speed up the aging process while also causing pathological phenomena, such as increased capillary permeability and impaired blood cell function [235]. All of these antioxidant systems behave differently depending on their structure and characteristics, whether hydrophilic or lipophilic, and where they are located (intracellular or extracellular, in cell or organelles membrane, in the cytoplasm, etc.). All of the above processes work in concert to establish a network that protects live cells from the damaging impacts of reactive oxygen species (ROS).

Figure 8 represents reactive oxygen species and neutralization with several biomolecules [236]. Hydrophobic amino acids in peptide chain contribute to their possible antioxidant effect [237]. Seaweeds also include nutraceutical and medicinal chemicals such phenols that have antioxidant activity. Polyphenols generated by seaweeds received special attention because their pharmacological action and broad range of health-promoting advantages, as polyphenols play a vital role in a variety of seaweed biological activities. Seaweed phenolic compounds are metabolites with hydroxylated aromatic rings that are chemically defined as molecules. In this context, Al-Amoudi et al. [25] stated that sulfated polysaccharides from three marine algae (Phaeophyta *Sargassum crassifolia* (S), Chlorophyta *Ulva lactuca* (U) and Rhodophyta *Digenea simplex* (D) exert antioxidant activity.

### 4.2. Antimicrobial Activity

Susceptibility testing of harmful microorganisms (e.g., bacteria and fungi) in the presence of possible compounds of interest is the focus of antimicrobial activity assays. Microbial infections can cause life-threatening illnesses, resulting in millions of deaths each year. Despite the fact that the discovery of penicillin pushed many aggressive pathogenic bacteria back, many strains evolved and developed remarkable resistance mechanisms to most antibiotics [238]. Variable solvents have different antibacterial action depending on their solubility and polarity. As a result, chemical compounds isolated from various seaweeds should be optimized for antibacterial activity by selecting the optimal solvent system [239]. Micro-algal cell-free extracts are already being studied as food and feed additives in an attempt to replace synthetic antibacterial chemicals currently in use. According to Tuney et al. [240], the antibacterial action of the extract is attributable to various chemical agents found in the extract, such as flavonoids, triterpenoids, and other phenolic compounds or free hydroxyl groups. Extraction procedures, solvents used, and the time window in which samples were collected all have the potential to alter antibacterial activity [241]. A variety of organic solvents had previously been recommended for screening algae for antibacterial activity.

Pérez et al. [242] demonstrated that seaweed extracts are effective at suppressing a variety of pathogens, including *E. coli* and *Salmonella*. The majority of the research looked at crude seaweed extracts of the chemicals in ethanol or methanol crude extract. It is unclear from these investigations whether the antibacterial activity is due to a single molecule or a combination of chemicals working together. Phytochemicals were shown in several investigations to produce significant bacterial cell-membrane damage by disrupting membrane integrity [243]. The active phytochemical substances can penetrate the bacterium after the membrane has been disrupted and interfere with DNA, RNA, protein, or polysaccharide formation, resulting in bacterial cell inactivation [244]. Two of the most common types of seaweeds, namely, the total phenolic, total flavonoid, and antibacterial properties of *Padina boryana* Thivy and *Enteromorpha* sp. marine algae were extensively examined, and the authors revealed that both seaweeds show antimicrobial activity against multiple pathogens [245].

### 4.3. Anticancer Activity

Cancers are life-threatening diseases that are considered to be a major public health issue around the world [246,247]. Uncontrolled cell development spreads into the surrounding tissues, resulting in the formation of a tumor mass [248]. Much research has looked into the anticancer potential of natural compounds derived from seaweeds, as well as the signaling pathways involved in anticancer activity [249]. Because those secondary metabolites have no hazardous effects, they have seen a lot of progress in the treatment of numerous diseases, including cancer. Thymoquinone (TQ) is one of the most important bioactive elements of black seeds, and it has been found to have numerous health advantages, including cancer prevention and treatment. Following on this, Algotiml et al. [250] studied the effect of biosynthesized Red Sea marine algal silver nanoparticles AgNPs on anticancer and antibacterial properties and the authors stated that due to their relatively moderate side effects, marine resources are currently being increasingly examined for antibacterial and anticancer medication prospects.

According to Palanisamy et al. [251], Fucoidans derived from Sargassum polycystum show antiproliferative characteristics at 50 g/mL. Additionally, Usoltseva et al. [252] also showed that native and deacetylated fucoidans (at 200 g/mL) from the brown seaweeds *Sargassum duplicatum*, *Sargassum feldmannii*, impeded colony formation in human colon cancer cells (DLD-1, HCT-116 or HT-29). According to findings of previous study [253], fucoidan extracted from the Brown seaweed Sargassum cinereum displays potent anticancer or apoptotic effects via preventing metastasis. In B-16 (mouse melanoma), CT-26 (murine colon cancer), HL-60 (human promyelocytic leukemia), or U-937 (human leukemic monocyte lymphoma) cell lines, polysaccharides produced through Pheophyceae *Ecklonia cava* show putative antiproliferative properties [254].

In addition, kappa-carrageenan extracted from *Hypnea musciformis* (Hm-SP) decreased proliferation of MCF-7 or SH-SY5Y cancer cell lines [255]. Additionally, polysaccharides derived from *Sargassum fusiforme* (SFPS) reduced SPC-A-1 cell proliferation in vitro and tumor formation in vivo [256]. Additionally, Ji and Ji [257] found that commercial laminaran (400–1600 g/mL) inhibited the growth of human colon cancer LoVo cells through stimulating mitochondrial or DR pathways. Additionally, Fucoidans isolated from *Undaria pinnatifida* have anticancer potential comparable to commercial fucoidans in cell lines Hela (human cervical), PC-3 (human prostate), HepG2 (human hepatocellular liver carcinoma), or A549 (carcinomic human alveolar basal epithelial) [258]. Moreover, previous study reported that fucoidan isolated from *Sargassum hemiphyllum* may increase miR-29b expression in human hepatocellular carcinoma cells, which aids in the lowering of DNA methyltransferase 3B expression [259]. Moreover, Fucoidans from *Fucus vesiculosus* were revealed to have anticancer potential, inducing apoptosis in MC3 human mucoepidermoid carcinoma cells via caspase-dependent apoptosis signaling cascade [260] (Figure 9).

### 4.4. Antidiabetics Activity

As a result of an unhealthy lifestyle, obesity, and stress, diabetes is becoming a global illness. Additionally, obesity has been on the rise in Saudi Arabia as a result of changing lifestyles and socioeconomic status [260,261]. There is a close association between obesity and type 2 diabetes. Drugs that suppress the enzymes α-glucosidase and α-amylase, which break down starch into glucose before it is absorbed into the bloodstream, could be used to treat diabetes [262]. It is necessary to look for effective therapeutic natural medications with less side effects. Garcimartn et al. [263] showed that a α-glucosidase inhibitory effect on restructured pork treated with seaweeds such as *Undaria pinnatifida*, *Himanthalia elongata*, and *Porphyra umbilicalis* caused a reduction in the blood glucose absorption. *Padina tetrastromatica* phenolic extracts inhibited both α-glucosidase and α-amylase, with higher inhibition linked with a higher phenolic concentration in the extracts. The extracts inhibited α-glucosidase (IC_50_ value of 28.8 g mL^−1^) and -amylase (IC_50_ value of 47.2 g mL^−1^) by 38.9 and 26.8%, respectively [264]. Similarly, α-glucosidase inhibitory action was observed in methanol, ethanol, and acetone extracts of *Durvillea antarctica*, methanol extracts of *Ulva* sp., and acetone extracts of Lessonia spicata [265]. Methanol extracts of *Padina tenuis* (400 µg mL^−1^) and ethanol extract of *Eucheuma denticulatum* (10 mg mL^−1^) and *Sargassum polycystum* (10 mg mL^−1^) significantly inhibited α-amylase by 60%, 67%, and 46%, respectively [266]. Recently, the acetone extract (80%) of brown seaweed Turbinaria decurrens was studied for its antihyperglycemic effects in alloxan induced diabetic wistar male rats [267]. The results showed a significant reduction in postprandial blood glucose levels of seaweed extracts treated rats to 180.33 mg dL^−1^ and 225.33 mg dL^−1^ at the dose of 300 mg/kg body weight and 150 mg/kg body weight, respectively, compared to diabetic control (565.0 mg dL^−1^) and positive control (115.33 mg dL^−1^). The bioactive compounds derived from algae and their application is illustrated in Table 9.

## 5. Seaweeds in Bio-Manufacturing Applications

Modern consumers are well aware of the nutritional value of food and the negative impact that synthetic preservatives may have worse effect on their health, so it is unsurprising that they prefer fresh and lightly preserved foods that are free of chemical preservatives, but contain natural compounds that may benefit their health [306].

### 5.1. Fertilizer and Soil Conditioners

Seaweed extracts have been frequently employed in agriculture in recent years to increase crop yield. This improvement is achieved by stimulating various physiological processes involved in plant growth and development, as well as improving final product quality (Figure 10). The use of traditional chemical fertilizers has expanded dramatically as result of world’s fast-growing population or ever-increasing food demand [307]. The usage of these chemical fertilizers, as well as their impacts, notably on environment, has become major source of worry [308]. As a result, farmers began to switch to organic farming rather than using synthetic agricultural fertilizers. Seaweeds are abundant or long-lasting resources discovered along the world’s coastlines, and they are important sources of food, feed, biofuels, cosmetics, fertilizers, nutraceuticals, and pharmaceuticals [309,310]. Due to their commercial importance or potential applications, seaweeds are used as fodder, cosmetics, human food, or biofertilizers [311]. Because of availability of various trace elements, vitamins, growth regulators, or amino acids, macroalgae extracts are currently being used as foliar sprays or presoaking for boosting growth or production of variety of plants, particularly crops [312]. Each year, more than 15 million tons of seaweed is produced, with much of it used as biofertilizers in agriculture or horticulture industries [313,314].

### 5.2. Medical and Pharmaceutical Use

#### 5.2.1. Biomedical Applications of Seaweeds

Bioactive chemicals found in seaweeds have features that make them appealing for biomedical applications. Many species of seaweeds have been employed in traditional medicine for a long time, notably in Asian nations, against goiter, nephritic disorders, anthelmintic, catarrh, and a few other ailments as medicaments or pharmaceutical auxiliaries, long before scientific study information [316]. *Fucus vesiculosus* has been used as a medicinal drug, primarily due to its iodine content, for obesity defects and goiters [316], for the treatment of sore knees [317], healing wounds [318], and also as herbal teas for their laxative effects [319]. The application of different seaweeds is presented in Table 10.

Chondrus crispus (Rhodophyta) carrageens have been used as mucilage against diarrhea, dysentery, gastric ulcers, and as a component of several health teas, such as for colds, for a long time. *Gelidium cartilagineum* (Rhodophyta) has been used in pediatric medicine in Japan for colds and scrofula [284]. *Ulva lactuca* (Chlorophyta) has been used for gout and as an astringent in folk medicine [284]. Rhodophyta extracts are very promising natural chemicals that could be used in biomedicine. Many species of Asian seaweeds are employed in traditional medicine, including *Gracilaria* spp. (Rhodophyta), which is used as a laxative, *Sargassum* spp. (Phaeophyceae), which is used to treat Chinese influence, and *Caloglossa* spp., *Codium* spp., *Dermonema* spp., and *Hypnea* spp. (Rhodophyta) [327].

Carrageenans’ biological actions make them attractive candidates for future antitumoral therapeutics since they activate antitumor immunity [328]. Kappaphycus species (Rhodophyta), for example, are used to treat ulcers, headaches, and tumors [327]. Antitumoral efficacy of carrageenans derived from Kappaphycus striatum against human nasopharyngeal carcinoma, human gastric carcinoma, and cervical cancer cell lines [329]. The bioactivity of chemicals from various Laurencia species (Rhodophyta) was investigated. In vitro, certain halogenated metabolites of *Laurencia papillosa* showed action against Jurkat (acute lymphoblastic leukemia) human tumor cells [330]. Laurencia obtuse extracts, specifically three sesquiterpenes, have been extracted and tested against Ehrlich ascites cancer cells. The sesquiterpenes were found to have antitumoral action against Ehrlich ascites cells [331]. *Gracilaria edulis* ethanol extracts showed antitumor efficacy in mice with ascites tumors [332].

*Undaria pinnatifida* (Phaeophyceae) has anti-inflammatory qualities and can be used to treat postpartum depression in women. This alga can also be used to treat edema and as a diuretic. Celikler et al. [333] investigated the antigenotoxic effect of *Ulva rigida* extracts in human cells in vitro (Chlorophyta).

Seaweeds have been suggested as a way to avoid neurogen-erative illnesses in investigations over the last decade [334]. Alzheimer’s disease (AD), Parkinson’s disease (PD), Huntington’s disease (HD), and Amyotrophic Lateral Sclerosis (ALS) are the most frequent [334]. According to Bauer et al., several studies highlighted the use of algal polysaccharides for the treatment of neurodegenerative illnesses [335]. Park et al. [336] found that mice treated with fucoidan extracts from Ecklonia cava had better memory and learning; consequently, the study implies favorable results in future human trials. In comparison to the control group, mice treated with polysaccharide isolated from *Sargassum fusiforme* demonstrated enhanced memory and cognition [337]. Dieckol and phlorofucofuroeckol, two phlorotannins from Ecklonia cava, are linked to an increase in the main central neurotransmitters in the brain, particularly Acetylcholine (ACh) [338]. Ahn et al. [339] investigated *Eisenia bicyclis* phlorotannins and found that 7-phloroeckol and phlorofucofuroeckol A were powerful neuroprotective agents against induced cytotoxicity, while eckol had a weaker impact.

#### 5.2.2. Pharmaceutical Applications of Seaweeds

Bioactive chemicals from seaweeds are used in the pharmaceutical industry to help develop new formulations for revolutionary treatments and to replace synthetic components with natural ones. Bioactive chemicals found in seaweeds have important pharmacological properties, including anticoagulant, antioxidant, antiproliferative, antititumoral, anti-inflammatory, and antiviral effects [340] (Table 11).

Fucoidans extracted from *Laminaria cichorioides* (Phaeophyceae) [351] and Fucus evanescens [352] behave like heparin in both in vitro and in vivo experiments, demonstrating anticoagulant activity by accelerating the development of antithrombin III to inhibit the effect against thrombin.

Fucoidans have a variety of characteristics. Pozharitskaya et al. [353] investigated the antioxidant, anti-inflammatory, anti-hyperglycaemic, and anticoagulant bioactivities of fucoidans isolated from *Fucus vesiculosus*. Even though their free-radical scavenging activity was lower than that of synthetic antioxidants, it was comparable to that of the natural antioxidant quercetin, which is found in plants. Furthermore, inhibition of both isoforms of the pro-inflammatory cyclooxygenase (COX-1) enzymes has been demonstrated, making fucoidans isolated from *Fucus vesiculosus* interesting substances for anti-inflammatory natural medicines [353]. Fucoidans from *Fucus vesiculosus* also have a role in fucoidan’s suppression of the enzyme DPP-IV. This enzyme is involved in the breakdown of incretin hormones, which prevents greater levels of glucose in the blood (postprandial hyperglycemia); a new pharmaceutical company is developing DPP-IV inhibitors to lower blood glucose levels and ensure anti-hyperglycaemic effects. As a result, according to Pozharitskaya et al. [353], fucoidans may be engaged in anti-hyperglycaemic activity via DPP-IV inhibition. *Sargassum fulvellum* (Phaeophyceae) has been found to contain a variety of bioactive compounds, including phlorotannins, grasshopper ketone, fucoidan, and polysaccharides, according to previous research. For years, *Sargassum fulvellum* extracts have been researched for their various pharmacological effects, including antioxidant, anticancer, anti-inflammatory, antibacterial, and anticoagulant properties [354].

Sargassum fulvellum extracts were studied for disorders such a lump, swelling, testicular discomfort, and urinary tract infections [355]. Agar made from red algae is frequently used in biomedicine as a suspension component in medicinal solutions and prescription goods, as well as anticoagulant and laxative agents in capsules [356]. The red algae *Gracilaria edulis* is well-known around the world for its biological and medicinal qualities. *Gracilaria edulis* extract exhibited antidiabetic, antioxidant, antibacterial, anticoagulant, anti-inflammatory, and antiproliferative characteristics [357]; consequently, these compounds could be used in new pharmaceutical formulations. Furthermore, Gunathilaka et al. [358] investigated the in vitro hypoglycemic efficacy of *Gracilaria edulis* phenolic, flavonoid, and alkaloid extracts. The suppression of carbohydrate-digesting enzymes, glucose absorption, and the generation of antiglycation end products demonstrated the red alga’s hypoglycemic potential. In vivo, *Ulva rigida* (Chlorophyta) has been shown to have a hypoglycemic impact [359].

Seaweeds’ antiviral qualities make them an excellent alternative for improving the health of infected persons; also, their use in pharmaceuticals will provide new and natural antiviral drugs that can replace synthetic chemicals. Furthermore, when compared to the creation of synthetic antivirals, the use of bioactive components from seaweeds is less expensive [360]. Antiviral activity of macroalgae has been discovered to protect against a variety of viruses, including HIV, Herpes Simplex Virus (HSV), genital warts [361], and hepatitis C (HCV) [362]. HSV [363], Encephalomyocarditis virus, Influenza “A” virus [364], and human metapneumovirus [365] are only a few of the viruses that Chlorophyta species have been shown to be effective against. The antiviral action of macroalgae is linked to a variety of substances such as as fatty acids and diterpenes, but most notably to the presence of Seaweed bioactive compounds [366], which can inhibit virus multiplication or help the immune system combat viral infection.

### 5.3. Cosmetic Industry

Cosmetics and cosmeceuticals are commonplace therapies for improving the skin’s appearance and treating several dermatological problems. Seaweeds are a valuable component in product development because of their wide range of functional, sensory, and biological properties. Consumer demand for green or eco-friendly products has risen in recent years. This pattern can be seen in the globally competitive cosmetics industry, in need of natural, secure, or effective ingredients to make innovative skin care products [367]. The usage of seaweed-isolated compounds in cosmetic products rose steadily as a result of various scientific studies revealing prospective skincare properties of seaweed bio-actives. Biologically active substances include carotenoids, polysaccharides, phlorotannins, fatty acids, sterols, tocopherol, vitamins, phycocyanins, or phycobilins [368,369,370,371,372]. In this context, a *Sargassum plagyophyllum* extract was shown to have antioxidant and anti-collagenase that can considered to be potent pharmaceutical ingredient for anti-wrinkle cosmetics action [373,374,375,376]. As a form of polyphenol, phlorotannins contain a group of heterogeneous polymeric molecules with substantial chemical modifications and various chemical structures [377]. These molecules can play a key role in the interaction between the skin and UVR, such as preventing radiation from penetrating the skin and lowering inflammation, oxidative stress, DNA damage, and maintaining signaling pathways intact. They also attracted a lot of interest because of their participation in several phototoxic pathways and mechanisms [378]. Brown algae *Sargassum fusiforme* [379], *Halidrys siliquosa* [380], *Padina australis* [381], *Sargassum coreanum* [382], and *Polycladia myrica* [383] have been explored for using in cosmetic products.

## 6. Materials and Methods

### Literature Search

The preferred reporting items for systematic reviews were used for the collection, identification, screening, selection, and analysis of the studies reviewed. A literature search was performed using different databases, including Scopus, Web of Science, Google Scholar, Wiley, MDPI, and PubMed. The search criteria included scientific articles on seaweeds published between 1989 and 2022. The keywords used in the literature search were “seaweeds” and “bioactivites OR “biological activities” OR “safety” OR “toxicity” OR “characteristics” OR “structure” OR “anticancer” OR “antidiabitics” OR “lipids” OR “polysaccarides” OR “phenolic compounds” OR “vitamines” OR “cosometics” OR “foods” OR “human” OR “minerals” OR “pigments and carotenoids” OR “protein” OR “amino acids”. The total number of articles found was 650. Studies focusing on the above keywords were selected, as well as those addressing the biological activity of seaweeds and the different applications of seaweeds. The figures were obtained from MDPI journals, and the chemical structure of compounds was designed by Chem windo 6 ver.4.1.1 Biorad edition.

## 7. Conclusions

Seaweeds include a wealth of bioactive compounds that could be used to develop novel functional ingredients for food as well as a therapy or prevention strategy for chronic diseases. Seaweeds could be an alternative source for synthetic substances that may help to increase consumer well-being via being incorporated into new functional foods or medications, as consumers have recently paid a lot of attention to natural bioactive compounds as functional ingredients in foods (Figure 11). However, because of the probable presence of hazardous pollutants such as heavy metals or their high iodine content, seaweed eating must be accompanied with an understanding of the hazards to human health. Because of the presence of numerous of innovative bioactive substances with potential anti-disease activities, using green extraction or purification processes of compounds from complex seaweed matrix is a viable or logical strategy for avoiding these health-related issues or creating added-value functional products.

## Figures and Tables

**Figure 1 marinedrugs-20-00342-f001:**
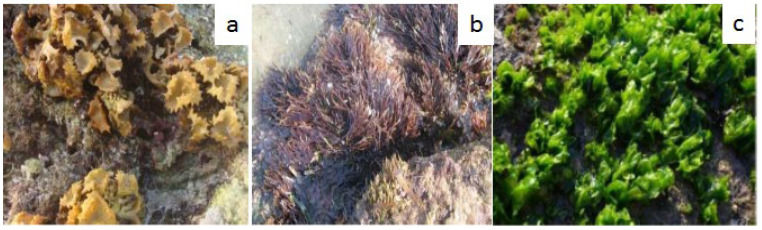
Three example species of brown (**a**) red (**b**) and green (**c**) seaweeds. Adapted from ref. [14] obtained from mdpi journals.

**Figure 2 marinedrugs-20-00342-f002:**
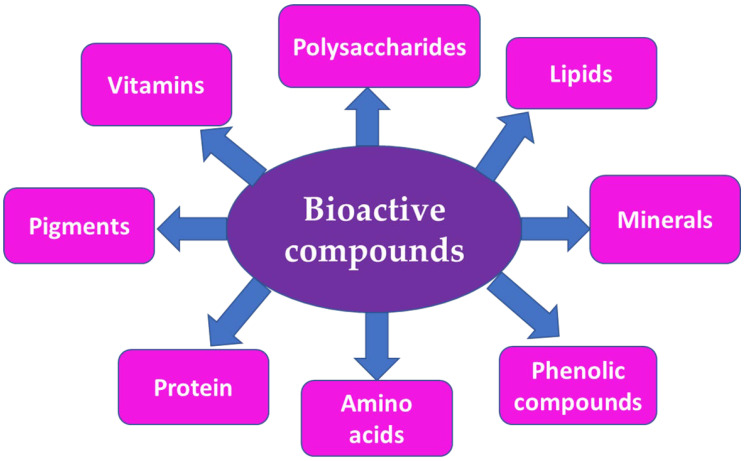
Main bioactive compounds from marine seaweeds.

**Figure 3 marinedrugs-20-00342-f003:**
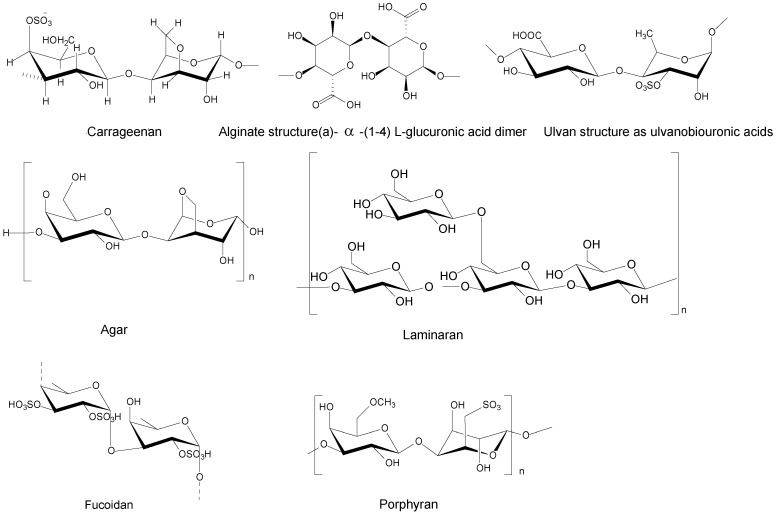
Chemical structures of different types of polysaccharides in seaweeds.

**Figure 4 marinedrugs-20-00342-f004:**
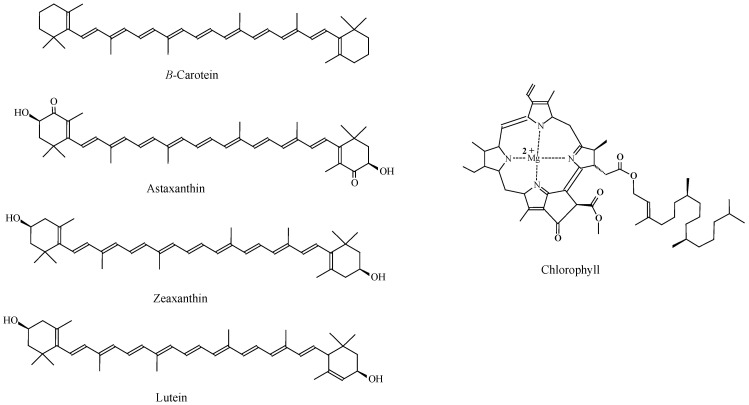
Chemical structures of different types of pigments in seaweeds.

**Figure 5 marinedrugs-20-00342-f005:**
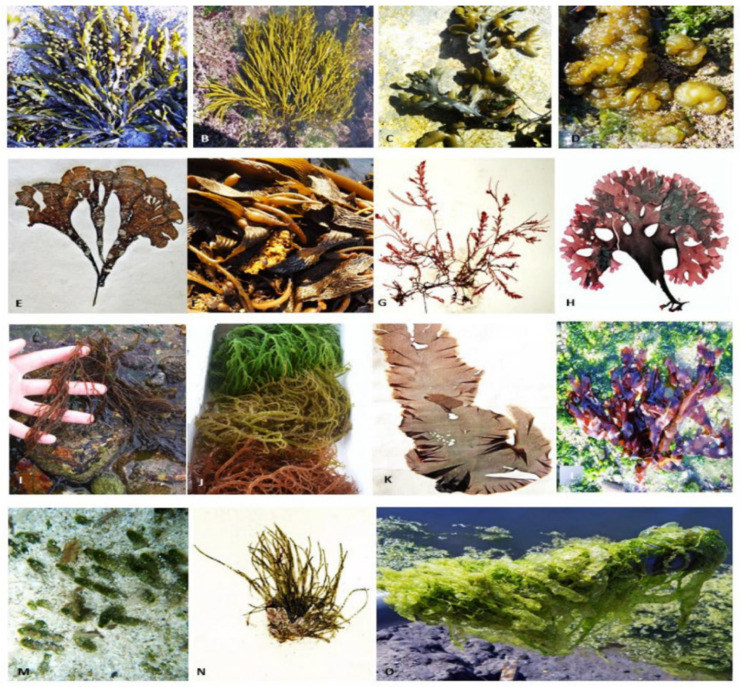
Several seaweeds synthesize phenolic substances. Adapted from ref [194] obtained from mdpi journals. (**A**)—*Ascophyllum nodosum* (P); (**B**)—*Bifurcaria bifurcata* (P); (**C**)—*Fucus vesiculosus* (P); (**D**)—*Leathesia marina* (P); (**E**)—*Lobophora variegata* (P); (**F**)—*Macrocystis pyrifera* (P); (**G**)—*Asparagopsis armata* (R); (**H**)—*Chondrus crispus* (R); (**I**)—*Gracilaria* sp. (R); (**J**)—*Kappaphycus alvarezii* (R); (**K**)—*Neopyropia* sp. (R); (**L**)—*Palmaria palmata* (R); (**M**)—*Dasycladus vermicularis* (Chl); (**N**)—*Derbesia tenuissima* (Chl); (**O**)—*Ulva intestinalis* (Chl); P—Phaeophyceae, R—Rhodophyta; Chl—Chlorophyta.

**Figure 6 marinedrugs-20-00342-f006:**
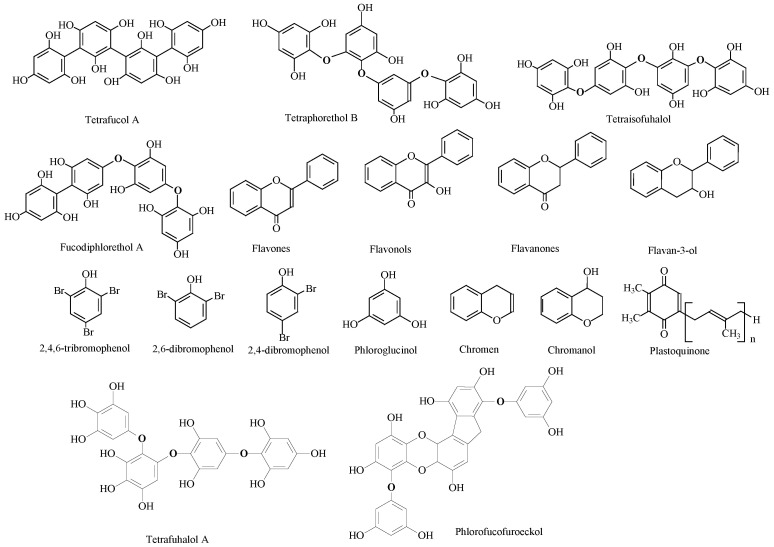
Chemical structures of different types of phenols in seaweeds.

**Figure 7 marinedrugs-20-00342-f007:**
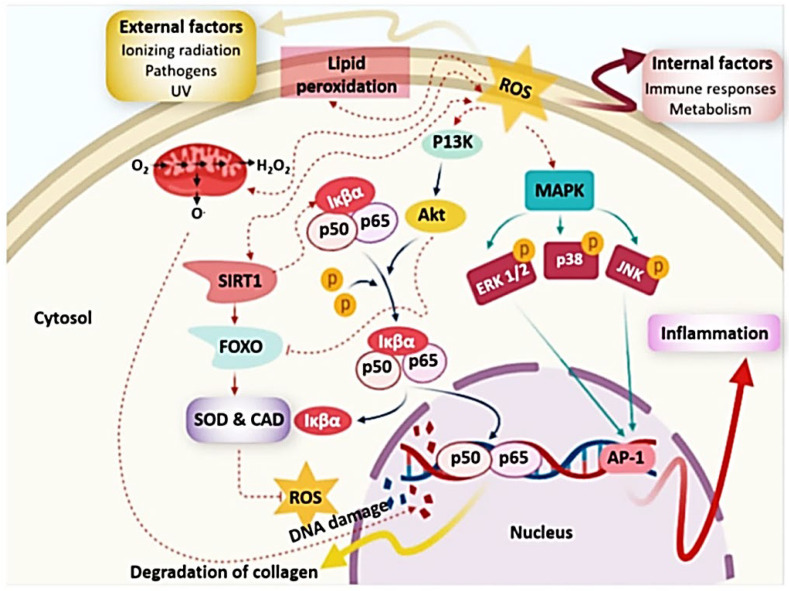
Damage caused via reactive oxygen species (ROS). Adapted from ref. [233] obtained from mdpi journals.

**Figure 8 marinedrugs-20-00342-f008:**
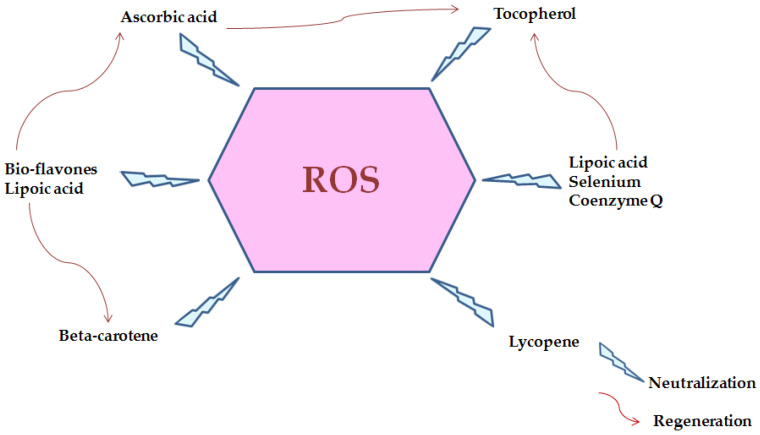
Reactive oxygen species and neutralization by several biomolecules.

**Figure 9 marinedrugs-20-00342-f009:**
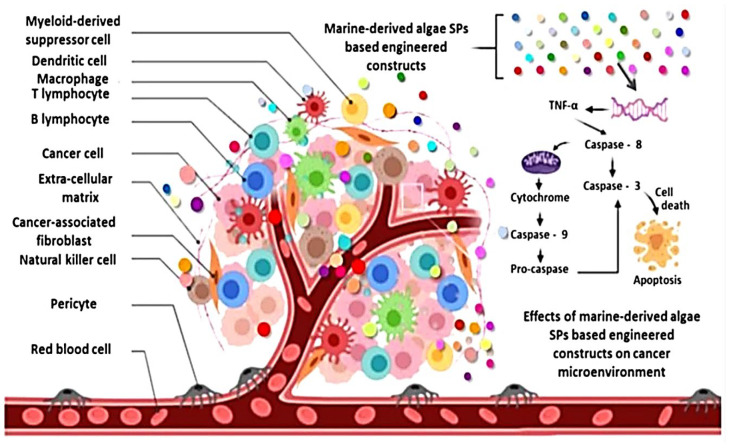
Demonstrate the ability of algal polysaccharide (SP)-based customized signals produced from sea algae to cause tumor cell death (apoptosis). Adapted from ref. [233] obtained from mdpi journals.

**Figure 10 marinedrugs-20-00342-f010:**
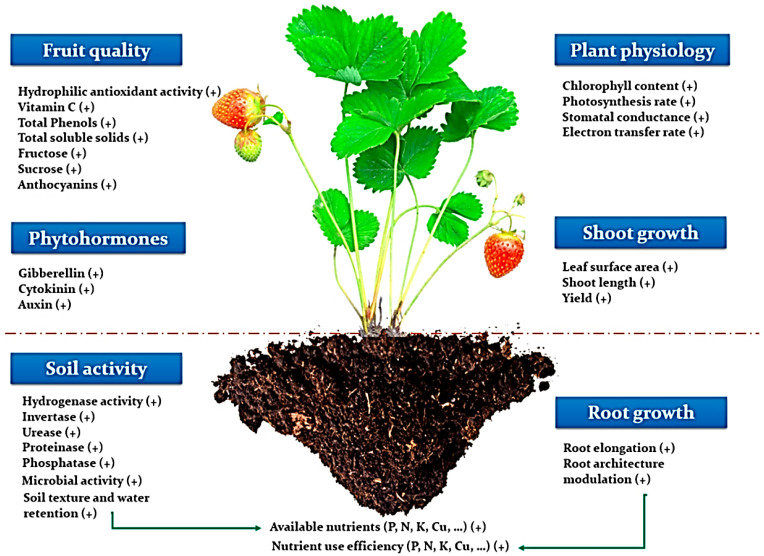
Illustration demonstrating beneficial effects of seaweed extracts on the entire soil-plant system. Such impacts include increased fruit quality and phytohormone content in plants, increased soil enzymatic activity, improved roots system, and overall physiological properties of plants. Adapted from ref. [315] obtained from mdpi journals.

**Figure 11 marinedrugs-20-00342-f011:**
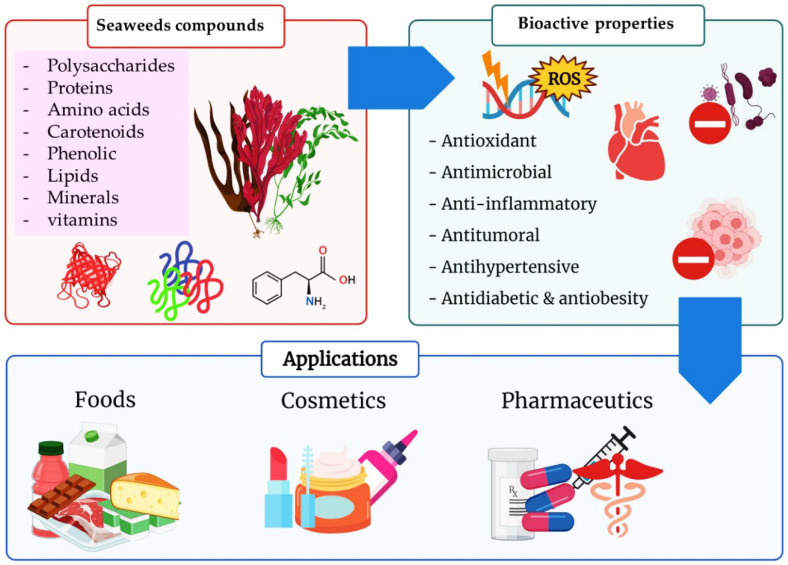
A summary for the bioactive compounds that have different biological activities and used in different applications. Adapted from ref. [384] obtained from mdpi journals.

**Table 1 marinedrugs-20-00342-t001:** Seaweeds polysaccharides and their roles in medicine.

Component	Species	Molecular Weight	Chemical Composition	Doses	Properties/Activities	References
Carrageenan	*Tribonema minus*	197 kDa	Heteropolysaccharide composed mainly of galactose	250 μg mL^−1^	Anticancer activity	[33]
Porphyran	*Chondrus armatus*	f 9.7–34.6 kDa	Mainly composed of 3,6-anhydro-L-galactose	327.3 μg mL^−1^	Anticancer activity	[34]
Fucoidans	*Cladosiphon okamuranus*	75.0 kDa	5.01 mg mL^−1^ of l-fucose, 2.02 mg mL^−1^ of uronic acids and 1.65 ppm of sulfate	1 mg mL^−1^	Anticancer activities	[35]
	*Fucus vesiculosus*	-	Fucose and Xylose	-	Antioxidant activity	[36]
Agar	*Gelidium amansii*	1.21 × 104 Da and 1.85 × 105 Da	(1–4)-linked 3,6-anhydro-α-L-galactose alternating with (1–3)-linked β-D-galactopyranose	25.6 mg L^−1^	Antioxidant activity	[37]
Laminaran	*Laminaria digitata*	-	β-(1,3)-glucan	10 µg mL^−1^	Antioxidant protection	[38]
Ulvan	*Ulva pertusa*	83.1 from 143.2 kDa	Rhamnose, and xylose	500 mg kg^−1^	Antioxidant and antihyperlipidemic activity	[39]
		143.47 kDa	Rhamnose and xylose	1.5 mg mL^−1^	Antioxidant Activity	[40]

**Table 2 marinedrugs-20-00342-t002:** Seaweeds polysaccharides and their roles in foods and cosmeceuticals.

Component	Species	Models	Doses	MW	Activity	Results	References
Carrageenan	*Padina tetrastromatic*	Paw edema in rats	20 mg kg^−1^	25 kDa	Anti-inflammation	COX-2 and iNOS inhibitions	[49]
Fucoidan	*Fucus vesiculosus*	Human malignant melanoma cells	100–400 µg mL^−1^	60 kDa	Anticancer activity	Inhibit cell proliferation	[50]
		B16 murine melanoma cells	550 µg mL^−1^	-	Anti-melanogenic	Inhibit tyrosinase and melanin	[51]
Ulvans	*Ulva* sp.	*Human dermal fibroblast*	100 and 500 µg mL^−1^	4–57 kDa	Anti-aging	Increase hyaluronan production	[52]
Laminaran	*Laminaria japonica*	In vitro	15 mg mL^−1^	250 kDa	antioxidant activity	ROS scavenging potential	[53]
Fucoidan	*Chnoospora minima*	RAW macrophages	27.82 µg mL^−1^	60 kDa	Anti-inflammation	Inhibition of LPS-induced NO production, iNOS, COX-2, and PGE2 levels	[54]
Fucoidan	*Sargassum* *hemiphyllum*	RAW 264.7 macrophage cells	dose-dependent manner	-	Anti-inflammation	Inhibit LPS-induced inflammatory response	[55]
Fucoidan	*Sargassum* *hemiphyllum*	B16 melanoma cells	dose-dependent manner	-	Anticancer	Activation of caspase-3	[56]

**Table 3 marinedrugs-20-00342-t003:** Different proteins accumulation of some seaweeds.

Seaweed	Species	Name of the Protein	Protein Yield %	References
*Ulva* sp.	Green algae	Glycoproteins (GP)	“UvGP-1” (0.54) “UvGP-2 DA”(0.52) “UvGP-2-DS”(1.98)	[81]
*Ulva lactuca*	Green algae	GP fraction G	ND	[82]
*Saccharina japonica*	Brown algae	Glycoprotein	0.27	[83]
*Solieria filiformis*	Red algae	Lectins “SfL-1” “SfL-2”	ND	[84]
*Solieria filiformis*	Red algae	Lectin “SfL”	ND	[85]
*Capsosiphon fulvescens*	Green algae	“Cf-hGP”	ND	[86]
*Undaria pinnatifida*	Brown algae	“UPGP”	ND	[87]

ND: Not detected; SfL: *Solieria filiformis* lectin; Cf-hGP: *Capsosiphon fulvescens* hydrophilic glycoproteins; UPGP: *Undaria pinnatifida* glycoprotein.

**Table 4 marinedrugs-20-00342-t004:** Amino acid composition accumulation of some seaweeds (g amino acid 100 g^−1^ protein).

No.	Amino Acids (AA)	*Caulerpa lentillifera* (Green Algae) [88]	*Ulva reticulate* (Green Algae) [88]	*Kappaphycus alvarezii* (Red Algae) [89]	*Gracilaria salicornia* (Red Algae) [89]	*Turbinaria ornata* (Brown Algae) [90]	*Durvillaea antarctica* (Brown Algae) [91]
	Essential AA						
1	Threonine	6.38	5.41	2.49	2.25	0.15	5.84
2	Valine	7.03	6.30	2.49	2.20	0.23	9.97
3	Lysine	6.63	6.02	1.51	-	0.20	4.22
4	Isoleucine	5.01	4.23	2.14	1.98	0.18	8.05
5	Leucine	8.00	7.90	2.34	2.16	0.26	15.88
6	Phenylalanine	4.93	5.26	2.11	1.79	0.19	9.97
7	Methionine	-	-	1.69	1.61	0.05	3.89
	Non essential AA						
8	Aspartic	11.56	12.50	3.33	-	0.53	4.17
9	Serine	6.14	6.39	2.68	2.90	0.10	5.38
10	Glutamic	14.39	12.98	11.67	2.79	0.58	17.87
11	Glycine	6.87	6.49	2.97	2.18	0.22	18.36
12	Arginine	7.03	8.65	2.40	2.40	0.19	4.83
13	Histidine	0.65	1.08	1.60	2.29	0.07	2.26
14	Alanine	6.87	8.09	2.93	2.51	0.23	9.57
15	Tyrosine	3.88	3.62	1.81	1.74	0.05	4.45
16	Proline	4.61	5.08	-	-	0.17	7.95
17	Cystin	-	-	-	-	0.00	0.78

**Table 5 marinedrugs-20-00342-t005:** Seaweeds proteins and their roles in medicinal.

Component	Properties/Activities	Seaweed	Doses	Molecular Weight	References
Peptide PPY1	Anti-aging	*Pyropia yezoensis*	250–1000 ng mL^−1^	532 Da	[100]
Peptides PYP1-5 and porphyra 334	Boost synthesis of elastin	*Porphyra yezoensis* f. *coreana Ueda*	0–200 μM	1622 kDa	[101]
Lactate and progerin	Reduce synthesis, anti-elastase, anti-collagenase	*Alaria esculenta*	-	112 KDa	[102]
Phycobiliproteins	Antioxidant	*Gracilaria gracilis*	0.5–30 mg mL^−1^	240 KDa	[103]
Deoxygadusol, palythene and usujirene	Antioxidant	*Rhodymenia pseudopalmata*	-	-	[104]
Palythine, palythinol, porphyra-334, asterina-330, shinorine, or usujirene	Antioxidant, antiproliferative	*Palmaria palmate*, *Mastocarpus stellatus*, *Chondrus crispus*	2.0–4.0 mg mL^−1^	244.24 KDa	[105]
Porphyra-334, shinorine, palythine and asterina-330	Antioxidant; UV-protective effect	*Gracilaria vermiculophylla*	-	346.33 KDa	[106]

**Table 6 marinedrugs-20-00342-t006:** Lipids accumulation of some seaweeds.

Seaweed	Species	Lipids g/100 g	EPA (%)	DHA (%)	References
*Caulerpa lentillifera*	Green algae	1.11 ± 0.05	0.86	-	[127]
*Codium fragile*	Green algae	1.5 ± 0.0	2.10 ± 0.00	-	[128]
*Ulva lactuca*	Green algae	1.27 ± 0.11	0.87 ± 0.16	0.8 ± 0.01	[129]
*Agarophyton chilense*	Red algae	1.3 ± 0.0	1.3 ± 0.01	-	[128]
*Porphyra*/*Pyropia* spp. (China)	Red algae	1.0 ± 0.2	10.4 ± 7.46	-	[128]
*Ascophyllum nodosum*	Brown algae	3.62 ± 0.17	7.24 ± 0.08	-	[130]
*Bifurcaria bifurcata*	Brown algae	6.54 ± 0.27	4.09 ± 0.08	11.10 ± 1.13	[130]
*Durvillaea antarctica*	Brown algae	0.8 ± 0.1	4.95 ± 0.11	1.66 ± 0.02	[129]
*Fucus vesiculosus*	Brown algae	3.75 ± 0.20	9.94 ± 0.14	-	[130]
*Himanthalia elongata*	Brown algae	<1.5	7.45	-	[131]
*Laminaria* spp.	Brown algae	1.0 ± 0.3	16.2 ± 8.9	-	[132]
*Macrocystis pyrifera*	Brown algae	0.7 ± 0.1	0.47 ± 0.01	-	[128]
*Sargassum fusiforme*	Brown algae	1.4 ± 0.1	42.4 ± 11.9	-	[132]
*Undaria pinnatifida*	Brown algae	4.5 ± 0.7	413.2 ± 0.66	-	[132]

EPA: eicosapentaenoic acid; DHA: docosahexaenoic acid.

**Table 7 marinedrugs-20-00342-t007:** The seaweeds lipids and their apllications.

Component	Molecular Mass	Properties/Activities	Seaweed	References
E-9-oxooctadec-10-enoic acid E-10-oxooctadec-8-enoic acid	282.46 g mol^−1^	Anti-inflammatory	*Gracilaria verrucosa*	[156]
Essential oil (tetradeconoic acid, hexadecanoic acid, (9Z)-hexadec-9-enoic acid) (9Z,12Z)-9,12-octadecadienoic acid	280.447 g mol^−1^	Antioxidant: radical scavenging Antibacterial activity upon *Staphylococcus aureus* and *Bacillus cereus*	*Laminaria japonica*	[157]
Fucosterol	412.69 g mol^−1^	Antioxidant: increased antioxidative enzymes (glutathione peroxidase, superoxide dismutase, catalase)	*Pelvetia siliquosa*	[158]
Fucosterol	412.69 g mol^−1^	Anti-inflammatory, Ati-photodamage: decreased UVB-induced MMPs	*Hizikia fusiformis*	[159]
Palmitic acid	256.430 g mol^−1^	Enzyme inhibition, Antioxidant	*Ulva rigida*, *Gracilaria* sp., *Saccharina latissima*, *Fucus vesiculosus*	[160]
Omega 3 fatty acids	909.4 g mol^−1^	Antioxidant	Brown algae	[161]
Arachidonic acid (ARA)	-	Improves growth and development of neonates	*P. purpureum*, *P. cruentum*	[162]
Eicosapentaenoic acid (EPA)	500 mg/day	Cognition, heart health, protection against arthrosclerosis, anti-inflammatory	*Nannochloropsis*, *P. tricornutum*, *P. cruentum*	[163,164]
Docosahexaenoic acid (DHA)	500 mg/day	Brain and eye health, cardiovascular benefits, nervous system development	*C. cohnii*, *Schizochytrium* sp., *Ulkenia* sp.	[162,163,164]
Fucosterol	1 and 10 μg mL^−1^	Anti-aging Inhibit MMP expression	*Hizikia fusiformis*	[165]
Polyunsaturated fatty acid	10.3 mg mL^−1^	Anti-inflammation	*Undaria pinnatifida*	[166]

**Table 8 marinedrugs-20-00342-t008:** Summarizes the key activities of carotenoids in human health.

Carotenoid	Seaweed Source	Effect	Model	Bioactive Concentration	Target	Reference
Astaxanthin	*Hematococcus pluvialis*	Antioxidant	Human monocytes (U-937)	10 μM	SHP-1	[168]
			Mice brain	2 mg/kg/day	MDA, NO, APOP, GSH.	[169]
			Leydig cells	10 μg/mL	StAR	[170]
		Antiproliferative	human prostatic adenocarcinoma (LNCaP)	10 μM	prostate specific antigen (PSA)	[171]
		immune system stimulation	transplantable methylcholanthrene-induced fibrosarcoma (Meth-A tumor)	40 mg/kg/day	interferon-g (IFN-γ)	[172]
		anti-obesity	Humans	0, 6, 12 and 18 mg/day	adiponectin	[173]
		Cardiovascular protective	spontaneously hypertensive rats (SHR)	50 mg/kg	blood pressure (BP)	[174]
Fucoxanthin	*Sargassum horneri*	antioxidant and protective	Vero cells	5, 50, 100 and 200 µM (50 µM H_2_O_2_)	DNA	[175]
		UV protection	Human fibroblasts	5, 50 and 100 µM (50 mJ/cm2 UV-B)	DNA	[176]
		Antioxidant	Retinol deficiency rats	0.83 µM	CAT, GST and Na^+^K^+^ATPase activity	[177]
		Antiproliferative	leukemia cells (HD-60)	11.3 and 45.2 μM	DNA fragmentation	[178]
			colorectal adenocarcinoma cells (Caco-2)	15.2 μM	DNA fragmentation	[178]
			colorectal adenocarcinoma cells (DLD-1)	15.2 μM	DNA fragmentation	[178]
			colorectal adenocarcinoma cells (CHT-29)	15.2 μM	DNA fragmentation	[178]
			human colorectal carcinoma (HCT116)	5 and 10 μM	Bcl-xL, PARP and caspase 3 and 7	[179]
		Antiproliferative	human urinary bladder cancer cells (EJ-1)	20 μM		[180]
		anti-obesity	Rats	2 mg	absorption of triglycerides, pancreatic lipase	[181]
Fucoxanthinol	*Corbicula fluminea*	Antiproliferative	human prostate cancer (PC-3)	2.0 μM	Bcl-xL, PARP and caspase 3 and 7	[179]
		anti-obesity	Rats	2 mg	absorption of triglycerides, pancreatic lipase	[181]
Halocynthiaxanthin	*Mastocarpus stellatus*	Antiproliferative	human neuroblastoma cells (GOTO)	5 μg/mL		[182]
β-carotene	*Kappaphycus alvarezii*	Antioxidant	Smokers	20 mg	Breath pentane	[183]
		Cure of erythema	Humans	30 to 90 mg/day		[184]
		Antiproliferative	murine osteosarcoma (LM8)	30 µM		[185]
		Antiinfiammatory	human umbilical vein endothelial cells (HUVECs)	0.02 µmol/L	VCAM-1, ICAM-1 and E-Selectin	[186]
Lutein	*Zostera noltii*	ADM prevention	Human Dermal Lymphatic Endothelial Cells (HLEC)	5 µM	DNA, lipid and protein level	[187]
		Cardiovascular protective	Human monocytes	0.1, 1, 10 and 100 nM	LDL associated with artery wall	[182]
Zeaxanthin	*Pyropia yezoensis*	ADM prevention	Human Dermal Lymphatic Endothelial Cells (HLEC)	5 µM	DNA, lipid and protein level	[187]

Abbreviations: SHP-1: protein tyrosine phosphatase non-receptor type 6; MDA: Malondialdehyde; NO: nitric oxide; APOP: protein oxidation product; GSH: glutathione; CAT: catalase; GST: glutathione S-transeferase; Bcl-xL: antiapoptotic factor; PARP: poly-ADP-ribose polymerase; (VCAM-1, ICAM-1): genes coding for vascular adhesion proteins.

**Table 9 marinedrugs-20-00342-t009:** Bioactive compounds derived from algae and their applications.

Algae Species	Bioactive Compound/Extract	Beneficial Activity	Mechanism of Action	Experimental Model	Reference
**Brown algae**					
*Ascophyllum nodosum*	Ascophyllan	Anticancer	Inhibit MMP expression	B16 melanoma cells	[268]
*Bifurcaria bifurcata*	Eleganonal	Antioxidant	DPPH inhibition	In vitro	[269]
*Chnoospora implexa*	Ethanol extract	Antimicrobial	Bacterial growth inhibition	*Staphylococcus aureus*, *Staphylococcus pyogenes*	[270]
*Chnoospora minima*	Fucoidan	Anti-inflammation	Inhibition of LPS-induced NO production, iNOS, COX-2, and PGE2 levels	RAW macrophages	[53]
*Cladosiphon okamuranus*	Fucoxanthin	Antioxidant	DPPH inhibition	In vitro	[271]
*Colpomenia sinuosa*	Ethanol extract	Antimicrobial	Bacterial growth inhibition	*S. aureus*, *S. pyogenes*	[270]
*Cystoseira barbata*	Fat-soluble vitamin and carotenoids	Antioxidant	High fat-soluble vitamin and carotenoid content	In vitro	[272]
*Dictyopteris delicatula*	Ethanol extract	Antimicrobial	Bacterial growth inhibition	*S. aureus*, *S. pyogenes*	[270]
*Dictyota dichotoma*	Algae extract	Antimicrobial	Inhibit the synthesis of the peptidoglycan layer of bacterial cell walls	*Penicillium purpurescens*, *Candida albicans*, *Aspergillus flavus*	[273]
*Eisenia arborea*	Phlorotannin	Anti-inflammation	Inhibit release of histamine	Rat basophile leukemia cells (RBL-2HE)	[274]
*Fucus evanescens*	Fucoidan	Anticancer	Inhibit cell proliferation	Human malignant melanoma cells	[50]
*Halopteris scoparia*	Ethanol extract	Anti-inflammation	COX-2 inhibition	COX inhibitory screening assay kit	[275]
*Laminaria japonica*	Fucoxanthin	Anti-melanogenic	Suppress tyrosinase activity	UVB-irradiated guinea pig	[276]
*Padina concrescens*	Ethanol extract	Antimicrobial	Bacterial growth inhibition	*S. aureus*, *S. pyogenes*	[270]
*Saccharina latissima*	Phenol	Antioxidant	High total phenolic content, DPPH scavenging activity and FRAP	In vitro	[277]
**Red algae**					
*Alsidium corallinum*	Methanol extract	Antimicrobial	Bacterial growth inhibition	*Escherichia coli*, *Klebsiella* *pneumoniae*, *Staphylococcus* *aureus*	[278]
*Ceramium rubrum*	Methanol extract	Antimicrobial	Bacterial growth inhibition	*Escherichia coli*, *Enterococcus faecalis*, *Staphylococcus aureus*	[278]
*Ganonema farinosum*	Ethanol extract	Antimicrobial	Bacterial growth inhibition	*S. aureus*, *S. pyogenes*	[270]
*Gelidium robustum*	Ethanol extract	Antimicrobial	Bacterial growth inhibition	*S. aureus*, *S. pyogenes*	[270]
*Jania rubens*	Glycosaminoglycan	Anti-aging	Collagen synthesis	Unknown	[279]
*Laurencia luzonensis*	Sesquiterpenes	Antimicrobial	Bacterial growth inhibition	*Bacillus megaterium*	[280]
*Palisada flagellifera*	Methanol extract	Antioxidant	β-carotene bleaching activity	In vitro	[281]
*Porphyra haitanensis*	Sulfated Polysaccharide	Antioxidant	ROS scavenging potential	Mice	[282]
*Schizymenia dubyi*	Phenol	Anti-melanogenic	Inhibit tyrosinase activity	In vitro	[283]
**Green algae**					
*Bryopsis plumose*	Polysaccharide	Antioxidant	ROS scavenging potential	In vitro	[54]
*Cladophora* sp.	Ethanol extract	Antimicrobial	Bacterial growth inhibition	*S. aureus*, *S. pyogenes*	[270]
*Entromorpha intestinalis*	Chloroform and methanol extract	Antioxidant	SOD activity is reduced	*Labidochromis caeruleus*	[284]
*Gayralia oxysperma*	Fucoxanthin	Antioxidant	High FRAP value (>6 μM/μg of extract)	In vitro	[285]
*Ulva dactilifera*	Ethanol extract	Antimicrobial	Bacterial growth inhibition	*S. aureus*, *Streptococcus* *pyogenes*	[270]
*Ulva fasciata*	Fucoxanthin	Antioxidant	DPPH inhibition (83.95%)	In vitro	[286]
*Ulva pertusa*	Polysaccharide	Antioxidant	ROS scavenging potential	In vitro	[54]
**Microalgae/Cyanobacteria**					
*Anabaena vaginicola*	Lycopene	Antioxidant Anti-aging	N/A	In vitro	[287]
*Arthrospira platensis*	Methanol extracts of exopolysaccharides	Antioxidant	N/A	In vitro	[287]
*Chlorella fusca*	Sporopollenin	Anti-aging	Protect cells from UV radiation	N/A	[288]
*Chlorella minutissima*	MAA	Anti-aging	Protect cells from UV radiation	N/A	[288]
*Chlorella sorokiniana*	MAA	Anti-aging	Protect cells from UV radiation	N/A	[288]
	Lutein	Anti-aging	Reduce UV induced damage	N/A	[289]
*Chlorella vulgaris*	Hot water extract	Anti-aging	Reduced activity of SOD	Human diploid fibroblast	[290]
		Anti-inflammation	Down-regulated mRNA expression levels of IL-4 and IFN-γ	NC/Nga mice	[291]
*Dunaliella salina*	β-carotene	Antioxidant	Protect against oxidative stress	Rat	[292]
	β-cryptoxanthin	Anti-inflammation	Reduced the production of IL-1β, IL-6, TNF-ɑ, the protein expression of iNOS and COX-2	LPS-stimulated RAW 264.7 cells	[293]
*Haematococcus* *pluvialis*	Astaxanthin (carotenoid)	Anti-aging	Inhibit MMP expression	Mice and human dermal fibroblasts	[294]
		Anticancer	ROS scavenging potential	Mice	[295]
*Nannochloropsis* *granulata*	Carotenoid	Antioxidant	DPPH inhibition	In vitro	[296]
*Nannochloropsis* *oculata*	Zeaxanthin	Anti-melanogenic	Inhibit tyrosinase	In vitro	[297]
*Nitzschia* sp.	Fucoxanthin	Antioxidant	Reduced oxidative stress	Human Glioma Cells	[298]
*Nostoc* sp.	MAA	Antioxidant	ROS scavenging potential	In vitro	[299]
*Odontella aurita*	EPA	Antioxidant	Reduce oxidative stress	Rat	[300]
*Planktochlorella* *nurekis*	Fatty acid	Antimicrobial	Bacterial growth inhibition	*Campylobacter jejuni*, *E. coli*, *Salmonella enterica* var.	[301]
*Porphyridium* sp.	Sulfated polysaccharide	Anti-inflammation Antioxidant	Inhibit proinflammatory modulator Inhibited oxidative damage	Unknown 3T3 cells	[282]
*Rhodella reticulata*	Sulfated polysaccharide	Antioxidant	ROS scavenging potential	In vitro	[282]
*Skeletonema marinoi*	Polyunsaturated aldehyde and fatty acid	Anticancer	Inhibit cell proliferation	Human melanoma cells (A2058)	[302]
*Spirulina platensis*	β-carotene and phycocyanin	Antioxidant Anti-inflammation	Inhibit lipid peroxidation Inhibit TNF-ɑ and IL-6 expressions	Mouse Human dermal fibroblast cells (CCD-986sk)	[303]
	Ethanol extract	Antimicrobial	Bacterial growth inhibition	*E. coli*, *Pseudomonas* *aeruginosa*, *Bacillus subtilis*, and *Aspergillus niger*	[304]
*Synechocystis* spp.	Fatty acids and phenols	Antimicrobial	Bacterial growth inhibition	*E. coli*, *S. aureus*	[305]

**Table 10 marinedrugs-20-00342-t010:** Biomedical effects of seaweed bioactive compounds.

Seaweed	Compound Extracted	Cell Lines/Animals Surveyed	Route of Administration	Dosage (μg/mL)	Effect	Reference
*Laminaria cichorioides* (Phaeophyceae)	Sulfated fucan	Human plasma	The lyophilized crude polysaccharide was dissolved in human plasma	10, 30, 50	In vitro anticoagulant activity	[320]
*Fucus evanescens* (Phaeophyceae)	Fucoidans	Human plasma Rat plasma	Intravenous Injection	125, 250, 500, 1000	In vitro and in vivo anticoagulant activity	[321]
*Gracilaria edulis* (Rhodophyceae)	Phenolic, Flavonoid and Alkaloid compounds	Bovine serum albumin (protein)	The extracts were tested on the protein	20, 40, 60, 80, 100, 120	Hypoglycemic activity	[322]
*Sargassum fulvellum* (Phaeophyceae)	Phlorotannins, grasshopper ketone, fucoidan and polysaccharides	Mice	Oral administration	Based on weight of mice	Antioxidant, anticancer, antiinflammatory, antibacterial, and anticoagulant activities	[323]
*Griffithsia* sp. (Rhodophyceae)	Griffithsin (protein)	MERS-CoV and SARS-CoV glycoproteins	The extracts were tested on the proteins	0.125, 0.25, 0.5, 1, 2	Antiviral activity against MERS-CoV virus and SARS-CoV glycoprotei	[324]
*Ulva rigida* (Chlorophyceae)	Ethanolic extract	Twenty-four male Wistar rats	Oral administration	500 mL of water with extracts in 2% wt/vol as drinking water for exposed groups per each day (from 3 to 30 days).	In vivo antihyperglycaemic, antioxidative and genotoxic/ antigenotoxic activities	[325]
*Saccharina japonica* (Phaeophyceae)	polysaccharides	SARS-CoV-2 S-protein	The extracts were tested on the proteins	50–500	In vitro inhibition to SARS-CoV-2	[326]

**Table 11 marinedrugs-20-00342-t011:** The potential pharmacological activity of brown, red and green seaweeds.

Component	Properties/Activities	Seaweed	Doses	Models	References
Fucoxanthins	Antitumoral activity on lung cancer cells	*Laminaria japonica*	12.5–100 μM	Female and male (1:1 ratio) BALB/c nude mice (18–20 g; 6–8 weeks of age)	[341]
	Antitumoral activity on MCF-7, HepG-2, HCT-116 cells	*Colpomenia sinuosa*, *Sargassum prismaticum*	100 and 200 mg/kg	Paracetamol-administered rats (one dose of 1 g/kg)	[342]
	Antitumoral activity on SiHa, Malme-3M cells	*Undaria pinnatifida*	1.5625, 6.25, 12.5, 25, 50, 80, 100 µM	Human cell lines	[343]
	Antimicrobial activity	*Cladosiphon okamuranus*	2–2000 µg/mL.	*Helicobacter pylori*	[344]
	Antimicrobial activity	*Laminaria japonica*	2, 3, 4, 5, 6, 7, and 7.5 mg/mL	*Staphylococcus aureus*, *Escherichia coli*	[345]
	Antimicrobial activity	*Fucus vesiculosus*	2, 4, 6, 8 and 10 mg/mL	*Staphylococcus aureus*, *Bacillus licheniformis*, *Escherichia coli*, *Staphylococcus epidermidis*	[346]
	Antiviral activity against ECHO-1, HIV-1, HSV-1, HSV-2	*Fucus evanescens*	200 μg/mL	Female outbred mice (16–20 g)	[347]
Sulfate polysaccharide	Antiviral activity against HSV-1, HVS-2	*Sargassum patens*	0.78–12.5 μg/mL	Vero cells (African green monkey kidney cell line)	[348]
	Anti-obesity, antidiabetic activities	*Gracilaria lemaneiformis*	5–10% Seaweed powder	Dawley laboratory rats (4 to 5 months old, 250–300 g)	[349]
Phloroglucinol	Anti-inflammatory activity	*Ecklonia cava*	1, 5, 10, 50 100 µM	HT1080 and RAW264.7 cells	[350]

## Data Availability

All data available in the review.

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
