# Peer review of "Phytochemical and Potential Properties of Seaweeds and Their Recent Applications: A Review"

_marinedrugs, 2022, doi:10.3390/md20060342_

Round 1
Reviewer 1 Report
I have read the manuscript and I have questions and recommendations.
1. There is no section "materials and methods" in the manuscript. The authors do not provide the information necessary for the review, search keywords, years of search, bases on which this search was conducted. Therefore, it is not clear what was the purpose of this review.
2. Specify whether the review is about wild or cultivated algae. Provide data in the "introduction" on the amount of algae consumed in various fields, especially in food, cosmetics and medicine.
3. Explain the need for drawing1. Is it just for displaying color?
4. In section 3, explain what "innovative food" is? Are algae not used in regular food? Or are they used in some unknown way?
5. Table 1 of section 3.1 must be supplemented the information on the use of polysaccharides in food, cosmetics, and also provide information in the table in more detail: molecular weight, composition, doses, models, real results. One of the main sources of fucoidan is Fucus vesiculosus, which grows in various regions (for example, https://doi.org/10.3390/md18050275, https://doi.org/10.3390/md20030193, etc.).
6. Please specify which "polysaccharide" you mean in the table with reference 41.
7. Table 2 of section 3.2 must be supplemented with information on the use of amino acids and proteins in food, cosmetics, and provide information in the table in more detail: molecular weight, structure, doses, models, real results. Discuss the results obtained. Compare the accumulation of amino acids and proteins in green, brown and red algae.
8. Table 3 of section 3.3 must be supplemented with information on the use of algae lipids in food, cosmetics, and also provide information in the table in more detail: molecular weight, structure, doses, models, real results. Discuss the results obtained. Compare lipid accumulation in green, brown and red algae. Decipher the composition of "Lipidic profile" and "Unsaturated fatty acids" in Table 3.
9. It is necessary to systematize the data of section 3.4 and supplement it with a table indicating the carotenoid, its purity, the source of the algae, dose / concentration, effects, reference drugs, literature references.
10. Phenolic compounds are one of the most important groups of algae metabolites, which have recently received increased attention. Please, in addition to photos of algae, consider phenolic compounds.
11. Section 3.6 expand and supplement with a table. Indicate the source, minerals, models, reference, aspect of application, literature reference.
12. Section 3.7 please systematize. Present data for lipid and water soluble vitamins. Compare the data for different types of algae, are there any patterns or differences in the accumulation of vitamins?
13. All sections 4 "Biological activity" should be supplemented with tables indicating metabolites, models, doses for in vivo and concentrations for in vitro / ex vivo, references, results obtained with numbers and literature. All data must be discussed and conclusions drawn.
14. Drawings must be designed in the same style.
15. Section 5.2 should be supplemented with information on medical use: substance, number of people, age, pathology, type of clinical trial, results. 16. Table 4 should be supplemented with information on models, in vivo doses, comparators, results obtained with numbers and references. Discuss the effect of molecular weight of monosaccharide composition on polysaccharides. Specify the name of phenols and phlorotannins.
16. Of particular interest is the use of phlorotannins for cosmetics (eg, https://doi.org/10.1007/s10811-022-02705-2). Discuss this aspect.
Author Response
Dear reviewer,
Thanks very much for your time and efforts for reviewing our manuscript. We are very grateful and appreciate your comments to improve the quality of the manuscript. We have carefully revised the manuscript according to the reviewer’s comments. Based on the suggestions, we have made an extensive modification on the revised manuscript. We have responded to the reviewers’ comments point by point and highlighted the revised parts in the manuscript. We wish this version is more suitable and cover all the comments.
Reviewer 1
I have read the manuscript and I have questions and recommendations.
1. There is no section "materials and methods" in the manuscript. The authors do not provide the information necessary for the review, search keywords, years of search, bases on which this search was conducted. Therefore, it is not clear what was the purpose of this review.
This is review article about seaweeds secondary metabolites and their applications so we not add materials and methods section and also not need to add year of search. This review aimed to study the bioactive compounds in seaweeds and the role of these compounds as antioxidant, anti-inflammatory, anti-cancer, antimicrobial and anti-diabetic activities. Also, there are many articles published in marine drugs journal and all mdpi journals did not included materials and methods for example:
https://doi.org/10.3390/md19050245
https://doi.org/10.3390/md18080384
Specify whether the review is about wild or cultivated algae. Provide data in the "introduction" on the amount of algae consumed in various fields, especially in food, cosmetics and medicine.
Done and we add it in line 70.
Explain the need for drawing1. Is it just for displaying color?
Yes, as an example to explain the diversity of photosynthetic pigments of algae.
In section 3, explain what "innovative food" is? Are algae not used in regular food? Or are they used in some unknown way?
We correct it, novel or regular food not innovative food
Table 1 of section 3.1 must be supplemented the information on the use of polysaccharides in food, cosmetics,
Done, some information you request were add in the table and others we wrote as paragraph and also we provide information in the table 1 in more detail: molecular weight, composition, doses, models, real results. One of the main sources of fucoidan is Fucus vesiculosus, which grows in various regions (for example, https://doi.org/10.3390/md18050275, https://doi.org/10.3390/md20030193, etc.).
Please specify which "polysaccharide" you mean in the table with reference 41.
Done and we delete it
Table 2 of section 3.2 must be supplemented with information on the use of amino acids and proteins in food, cosmetics,
Done and we add it
provide information in the table in more detail: molecular weight, structure, doses, models, real results. Discuss the results obtained.
We done the best to cover this point
Compare the accumulation of amino acids and proteins in green, brown and red algae.
Done
Table 3 of section 3.3 must be supplemented with information on the use of algae lipids in food, cosmetics,
Done we add all the information
provide information in the table in more detail: molecular weight, structure, doses, models, real results. Discuss the results obtained.
Dear doctor very thanks to you for all comments we try to add all information’s you need
Compare lipid accumulation in green, brown and red algae.
Done
Decipher the composition of "Lipidic profile" and "Unsaturated fatty acids" in Table 3.
Done
It is necessary to systematize the data of section 3.4 and supplement it with a table indicating the carotenoid, its purity, the source of the algae, dose / concentration, effects, reference drugs, literature references.
Done
Phenolic compounds are one of the most important groups of algae metabolites, which have recently received increased attention. Please, in addition to photos of algae, consider phenolic compounds.
Done we add figure about the structure of the phenolic compound and information about phenolic compound in different seawweds.
Section 3.6 expand and supplement with a table. Indicate the source, minerals, models, reference, aspect of application, literature reference.
Done we add several details about minerals in seaweeds and comparison between minerals content in different microalgae.
- Section 3.7 please systematize. Present data for lipid and water soluble vitamins. Compare the data for different types of algae, are there any patterns or differences in the accumulation of vitamins?
Done
All sections 4 "Biological activity" should be supplemented with tables indicating metabolites, models, doses for in vivo and concentrations for in vitro / ex vivo, references, results obtained with numbers and literature. All data must be discussed and conclusions drawn.
Done
Drawings must be designed in the same style.
Drawing obtained from mdpi journal to help in the explain of some items in the review
Section 5.2 should be supplemented with information on medical use: substance, number of people, age, pathology, type of clinical trial, results.
Done we add more information about Medical and pharmaceutical uses
- Table 4 should be supplemented with information on models, in vivo doses, comparators, results obtained with numbers and references. Discuss the effect of molecular weight of monosaccharide composition on polysaccharides.
Done
Specify the name of phenols and phlorotannins.
Done
Of particular interest is the use of phlorotannins for cosmetics (eg, https://doi.org/10.1007/s10811-022-02705-2). Discuss this aspect.
Done
Reviewer 2 Report
This review presents current knowledge on the phytochemical properties of seaweeds and their potential properties and recent applications. The authors review a wide variety of articles and present meaningful content to the reader. Therefore, I believe that this paper deserves to be published in Marine Drugs, but the some points need to be added and revised.
Major
- In Tables 1-4, information on content, evaluation model, effective dose, and bioavailability should be presented.
- Figures 1-8 are all photographs and conceptual drawings, with no information on the structure of the bioactive substance.
Minor
Ln23: “… phenols polysaccharides, and sterols,…” should be “… phenols, polysaccharides, and sterols,…”.
Author Response
Dear reviewer,
Thanks very much for your time and efforts for reviewing our manuscript. We are very grateful and appreciate your comments to improve the quality of the manuscript. We have carefully revised the manuscript according to the reviewer’s comments. Based on the suggestions, we have made an extensive modification on the revised manuscript. We have responded to the reviewers’ comments point by point and highlighted the revised parts in the manuscript. We wish this version is more suitable and cover all the comments.
Reviewer 2
this review presents current knowledge on the phytochemical properties of seaweeds and their potential properties and recent applications. The authors review a wide variety of articles and present meaningful content to the reader. Therefore, I believe that this paper deserves to be published in Marine Drugs, but the some points need to be added and revised.
Major
- In Tables 1-4, information on content, evaluation model, effective dose, and bioavailability should be presented.
Done we add the information you wanted in the tables
- Figures 1-8 are all photographs and conceptual drawings, with no information on the structure of the bioactive substance.
- Done we add figures about the structure of bioactive compounds
Minor
Ln23: “… phenols polysaccharides, and sterols,…” should be “… phenols, polysaccharides, and sterols,…”.
Done
Round 2
Reviewer 1 Report
I have read the revised manuscript and I have questions again.
- There is no section "materials and methods" in the manuscript. The authors do not provide the information necessary for the review, search keywords, years of search, bases on which this search was conducted. Therefore, it is not clear what was the purpose of this review.
- In section 4.4, explain what are “innovative mechanisms of action” for antidiabetic activity? Provide literary references.
- Table 1 of section 3.1 must be supplemented with information on the use of polysaccharides in food, cosmetics, and also provide information in the table in more detail: doses, models, real results.
- Reference 63 does not contain information about fucoidan with M.m. 34.4 kDa. Provide correct data, please.
-
Compare the accumulation of amino acids and proteins in green, brown and red algae.
- Table 3 of section 3.3 must be supplemented with information on the use of algae lipids in food, cosmetics, and also provide information in the table in more detail: doses, models, real results. Discuss specific data in numbers. Compare lipid accumulation in green, brown and red algae.
- Please indicate the literary reference for figure 7.
- Reference 60 deals with algal polysaccharides but not polyphenols (section 5.3). Provide correct information.
-
Figures must be designed in the same style.
- Literary reference 346 deals with polysaccharides, and you write about polyphenols. Provide correct information.
- One source was cited twice (63 and 311). Provide correct information.
Author Response
Dear reviewer,
Thanks very much for your time and efforts for reviewing our manuscript. We are very grateful and appreciate your comments to improve the quality of the manuscript. We have carefully revised the manuscript according to the reviewer’s comments. Based on the suggestions, we have made an extensive modification on the revised manuscript. We have responded to the reviewers’ comments point by point and highlighted the revised parts in the manuscript. We wish this version is more suitable and cover all the comments.
I have read the revised manuscript and I have questions again.
- There is no section "materials and methods" in the manuscript. The authors do not provide the information necessary for the review, search keywords, years of search, bases on which this search was conducted. Therefore, it is not clear what was the purpose of this review.
Done
- In section 4.4, explain what are “innovative mechanisms of action” for antidiabetic activity? Provide literary references.
Done we add literary references to this part and delete this sentence
- Table 1 of section 3.1 must be supplemented with information on the use of polysaccharides in food, cosmetics, and also provide information in the table in more detail: doses, models, real results.
Done we add it in separate table
- Reference 63 does not contain information about fucoidan with M.m. 34.4 kDa. Provide correct data, please.
Done we corrected it and add the correct MW and correct reference
- Compare the accumulation of amino acids and proteins in green, brown and red algae.
Done we add table about the comparison
- Table 3 of section 3.3 must be supplemented with information on the use of algae lipids in food, cosmetics, and also provide information in the table in more detail: doses, models, real results. Discuss specific data in numbers.
Done, we try to add all the points
- Compare lipid accumulation in green, brown and red algae.
Done we add table about the comparison
- Please indicate the literary reference for figure 7.
Done
- Reference 60 deals with algal polysaccharides but not polyphenols (section 5.3). Provide correct information.
Very thanks to you we delete it from text
- Figures must be designed in the same style.
Done we tried to make the figures as the same.
- Literary reference 346 deals with polysaccharides, and you write about polyphenols. Provide correct information.
Done we correct the reference
- One source was cited twice (63 and 311). Provide correct information.
Done
Reviewer 2 Report
Because the authors revised their manuscript sincerely, I think this manuscript should be considered for publication.
Author Response
Dear reviewer 2,
Thanks very much for your time and efforts for reviewing our manuscript. We are very grateful and appreciate your comments to improve the quality of the manuscript.